# Structural landscape of activation, desensitization and inhibition in the human TRPM4 channel

Celso M. Teixeira-Duarte[1,2], Weizhong Zeng[1,2] & Youxing Jiang ®[1,2]✉

TRPM4 is a member of the transient receptor potential melastatin channel subfamily and functions as a $Ca^{2+}$-activated monovalent-selective cation channel. It is widely expressed in various cells and tissues, where its activation depolarizes the plasma membrane potential and modulates various $Ca^{2+}$-dependent biological processes. TRPM4 activity is potentiated by membrane phosphatidylinositol 4,5-bisphosphate ($PtdIns(4,5)P_2$) and inhibited by cytosolic free adenosine triphosphate (ATP), allowing the channel to transition between different functional states in response to dynamic changes in cellular $Ca^{2+}$, ATP and $PtdIns(4,5)P_2$ levels during signaling events. Here we present single-particle cryo-electron microscopy structures of human TRPM4 in four distinct states: apo closed, $Ca^{2+}$-bound putative desensitized, $Ca^{2+}$-$PtdIns(4,5)P_2$-bound open and ATP-bound inhibited. Combined with mutagenesis and electrophysiological analyses, these structures reveal the molecular mechanisms underlying TRPM4 activation, desensitization and inhibition. Given the central roles of $Ca^{2+}$, $PtdIns(4,5)P_2$ and ATP in cellular signaling, this work provides a structural foundation to decipher the physiological functions of TRPM4 across diverse biological systems.

The $Ca^{2+}$-activated monovalent-selective TRPM4 cation channel is a member of the transient receptor potential melastatin channel subfamily[1–3]. Comprising the largest subfamily of tetrameric TRP channels, TRPM channels share four signature intracellular N-terminal melastatin homology region domains (MHR1–MHR4) followed by S1–S6 transmembrane domains, the TRP domain, a rib helix and a coiled-coil helix that forms a four-helix bundle in a channel tetramer. Some family members also contain an additional enzyme domain at their C terminus[4]. Among the eight members of the TRPM subfamily, TRPM4 and TRPM5 share high sequence identity (~50% identity) and similar biophysical properties, including $Ca^{2+}$ activation and monovalent cation conduction[1,3,5].

While TRPM5 is mainly found in taste receptor cells[6,7], TRPM4 is widely expressed in the brain, pancreas, kidney and heart, as well as in immune cells and the central nervous system[1,3,8–10]. Therefore,

TRPM4 has important roles in a myriad of physiological processes such as insulin secretion[11], immune response[12], cell death[13,14] and cardiac conduction[10,15]. *TRPM4* mutations are associated with various cardiac dysfunctions including atrioventricular conduction block[16], progressive familial heart block type I[17], Brugada syndrome[18] and long QT syndrome[19]. Notably, TRPM4 has been shown to directly regulate the necrosis process by mediating sodium entry inside the cell[13,14,20]. The sodium accumulation then leads to water entry and subsequent cell swelling and bursting[14,20]. As TRPM4 upregulation has been detected in several cancer cells[21], it has become the center of emerging cancer therapies by promoting necrotic cell death in cancers[14,20].

TRPM4 is activated by cytosolic $Ca^{2+}$ and modulated by phosphatidylinositol 4,5-bisphosphate ($PtdIns(4,5)P_2$) and adenosine triphosphate (ATP)[22–24]. $PtdIns(4,5)P_2$ regulates TRPM4 by potentiating the $Ca^{2+}$ activation of the channel. The reduction in $PtdIns(4,5)P_2$ level in

[1]Howard Hughes Medical Institute and Department of Physiology, University of Texas Southwestern Medical Center, Dallas, TX, USA. [2]Department of Biophysics, University of Texas Southwestern Medical Center, Dallas, TX, USA. ✉e-mail: youxing.jiang@utsouthwestern.edu

the plasma membrane can result in channel desensitization[22,24]. Interestingly, while free cytosolic ATP acts as a direct endogenous TRPM4 inhibitor, $Mg^{2+}$-chelated ATP was shown to increase channel activity by alleviating channel desensitization[23–26]. This channel recovery from desensitization upon exposure to $Mg^{2+}$-ATP is likely because of the replenishment of membrane $PtdIns(4,5)P_2$ by the ATP activation of lipid kinases[24].

Several TRPM4 structures in the closed state have been determined previously, revealing the overall architecture of the channel and the binding sites for calcium and ATP[25,27–29]. The structure of TRPM4 in an open conformation was determined in a recent study by exposing the channel to a high temperature (37 °C) in the presence of $Ca^{2+}$ and an exogenous modulator, decavanadate (DVT)[30], likely because the high temperature potentiates the $Ca^{2+}$ activation and DVT modulation. However, several fundamental questions related to TRPM4 regulation by endogenous cellular stimuli remain elusive, such as where $PtdIns(4,5)P_2$ binds, how $PtdIns(4,5)P_2$ potentiates channel activity, what is the structural basis of TRPM4 desensitization and how free ATP inhibits the channel. In this study, we present single-particle cryo-electron microscopy (cryo-EM) structures of the human TRPM4 channel in different states, including apo closed, $Ca^{2+}$-bound putative desensitized, $Ca^{2+}$-$PtdIns(4,5)P_2$-bound open and ATP-bound inhibited states. These structures encompass a conformational landscape of TRPM4 activation, desensitization and inhibition, providing fundamental insights into the structural mechanisms of TRPM4 regulation in response to physiologically important signaling molecules.

## Results

### Functional and structural characterization of human TRPM4 with native ligands

Patch-clamp recordings were used to characterize the $Ca^{2+}$ and $PtdIns(4,5)P_2$ activation of TRPM4 (Methods). Water-soluble short-chain synthetic $PtdIns(4,5)P_2$ diC8 can effectively potentiate the $Ca^{2+}$ activation of TRPM4 (refs. 22,24,25) and was used as the native $PtdIns(4,5)P_2$ substituent in most recordings. Figure 1a–c illustrate the key electrophysiological features of human TRPM4 activated by $Ca^{2+}$ and $PtdIns(4,5)P_2$. Upon its initial activation by cytosolic $Ca^{2+}$, TRPM4 quickly desensitizes to a steady-state level with lower channel activity in response to $Ca^{2+}$. The desensitized TRPM4 exhibits voltage-dependent gating with relatively higher channel open probability at depolarizing membrane potential. This decrease in channel activity is caused by the loss of $PtdIns(4,5)P_2$ in the membrane because of $Ca^{2+}$-induced phospholipase C activation[22,24]. While $PtdIns(4,5)P_2$ alone cannot activate TRPM4, it functions as a positive modulator whose presence alongside $Ca^{2+}$ reverses the channel desensitization and restores the TRPM4 currents to the initial $Ca^{2+}$-activated level. $Ca^{2+}$ activation in the presence of $PtdIns(4,5)P_2$ also mitigates the voltage dependence of the channel[22,24,25]. These functional data indicate a requirement for the simultaneous binding of calcium and $PtdIns(4,5)P_2$ to stabilize TRPM4 in a fully open state. As native porcine brain $PtdIns(4,5)P_2$ was used in our structural study, we also tested its potentiation effect on TRPM4 (Extended Data Fig. 1). The long-chain native $PtdIns(4,5)P_2$ is insoluble and forms liposomes in solution, making it difficult to fuse into the patched membrane. Before recording, the liposome-containing bath solution was sonicated to improve the efficiency of lipid vesicle fusion into the patched membrane. With sufficient perfusion time, the native $PtdIns(4,5)P_2$ also potentiates the TRPM4 activity in the presence of $Ca^{2+}$ and reverses the channel desensitization similar to the short-chain $PtdIns(4,5)P_2$ diC8 (Extended Data Fig. 1).

Aiming to reveal the structural basis of TRPM4 activation by its native ligands, we purified detergent-solubilized human (h)TRPM4 in the presence of porcine brain $PtdIns(4,5)P_2$ (Methods). $Ca^{2+}$ was added to the protein sample to activate the channel before the EM grid preparation and data collection. The single particles used for final structural determination were partitioned into three classes (Extended Data Fig. 2 and Table 1). The first class, with about 46% of the particles, yielded a

2.8 Å structure of TRPM4 in the apo closed conformation with no bound ligands. The second class, with about 20% of the particles, yielded a 2.5 Å structure of TRPM4 in a fully open state with both $Ca^{2+}$ and $PtdIns(4,5)P_2$ bound. The third class, with about 34% of the particles, yielded a 3.2 Å structure of TRPM4 with bound $Ca^{2+}$ but the pore remained closed, representing a putative desensitized state, as discussed later. These structures, along with ATP-inhibited hTRPM4 presented later, illustrate the structural landscape of hTRPM4 activation, desensitization and inhibition (Figs. 1d,e and Extended Data Figs. 2 and 3).

### Calcium and $PtdIns(4,5)P_2$ binding in hTRPM4

The bound ligands $Ca^{2+}$ and $PtdIns(4,5)P_2$ can be unambiguously identified from the EM density map of the open hTRPM4 structure (Fig. 2a). Two bound $Ca^{2+}$ ions are observed within each subunit. The first $Ca^{2+}$ site is located in a solvent-exposed pocket within the transmembrane S1–S4 domains and the bound $Ca^{2+}$ is chelated by the side chains of E828 and Q831 from S2 and N865 and D868 from S3 (Fig. 2b). E1068 on TRP helix 1 was previously shown to be important for $Ca^{2+}$ activation[31]. While not directly chelating $Ca^{2+}$, E1068 is positioned at the bottom of the pocket with its side chain extended upward close to the bound $Ca^{2+}$. Salt-bridged with D868, the side chain of R905 at the top of the pocket is oriented and extended downward toward E1068 upon $Ca^{2+}$ binding. The R905 side chain, along with the bound $Ca^{2+}$, likely engages in an electrostatic interaction with E1068 to facilitate channel activation. Indeed, both E1068A and R905A substitutions have profound effects on $Ca^{2+}$ activation (Fig. 2c and Extended Data Fig. 4a). While $Ca^{2+}$ binding stabilizes the side chains of the surrounding residues at the pocket, it also induces some local conformational changes that trigger the opening of the pore. As further discussed in the next section, the $Ca^{2+}$-induced local movement is quite subtle but can be propagated into a much larger movement on other parts of the channel, most notably the cytosolic domain.

Intriguingly, mutagenesis of those surrounding residues at this $Ca^{2+}$-binding pocket yielded two opposing effects on $Ca^{2+}$ activation (Fig. 2c and Extended Data Fig. 4a). While D868A, R905A and E1068A substitutions mitigate $Ca^{2+}$ activation of the channel in the steady state, E828A and N865A represent two gain-of-function (GOF) substitutions that facilitate the $Ca^{2+}$ activation of the TRPM4 channel, with the N865A mutant having a particularly potent GOF effect. It is unclear how these gain-of-function substitutions potentiate $Ca^{2+}$ activation. With multiple charged residues congregated at the $Ca^{2+}$-binding pocket, we speculate that the two GOF substitutions may allow for a rearrangement of $Ca^{2+}$ coordination that facilitates rather than hinders the $Ca^{2+}$-induced conformational change for channel activation.

The second bound $Ca^{2+}$ is observed in the intracellular MHR domains and is coordinated by D270 from MHR2 and C385, D395 and E396 from MHR3 (Fig. 2d). This intracellular $Ca^{2+}$ site was initially identified in TRPM5 and was also observed in the recent open TRPM4 structure obtained at 37 °C (refs. 30,32). In the closed state, however, D270 and C385 are far apart from D395 and E396 because of the conformational changes at the intracellular domains, suggesting that this $Ca^{2+}$ site only exists in the open state. This would imply that the intracellular $Ca^{2+}$ binding may help stabilize the channel in the open conformation but is not essential for initial channel activation. Interestingly, among the three $Ca^{2+}$-chelating acidic residues, only the E396 substitution impacts the channel activity (Fig. 2e and Extended Data Fig. 4b).

$PtdIns(4,5)P_2$ binds beside the S1–S4 domain at the junction formed by the S3 and S4 helices, the S4–S5 linker and TRP helix 1 (Fig. 2f). One of its acyl chains is well resolved and runs parallel along the S3 and S4 helices with extensive hydrophobic contact. The inositol 1,4,5-trisphosphate head group is positioned on the cytosolic side and defines the key ligand–protein interactions with residues from various parts of the channel, including R664 from the MHR4 domain, S924 and K925 at the C terminus of the S4–S5 linker, K928 at the N terminus of S5 and Y1057 and Q1061 on TRP helix 1 (Fig. 2f). Substitutions of those positively charged residues that predominantly interact with the

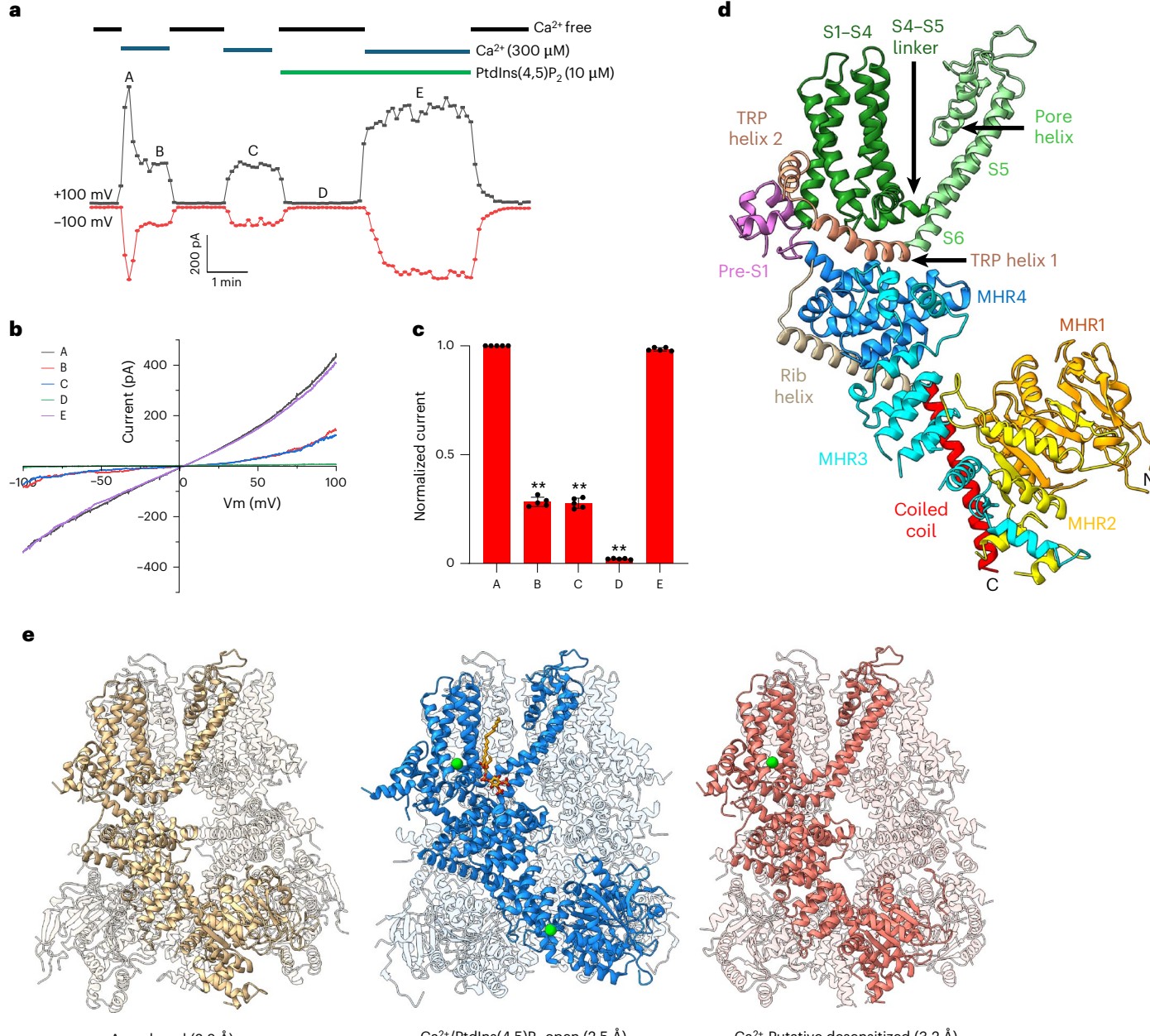

**Fig. 1 | Ligand activation and overall structures of hTRPM4. a**, Macroscopic currents of TRPM4 overexpressed in HEK293 cells at ±100 mV in an inside-out patch with the presence or absence of $Ca^{2+}$ and $PtdIns(4,5)P_2$ in the bath (cytosolic). A–E mark various functional states of TRPM4: A, initial $Ca^{2+}$-activated state before $PtdIns(4,5)P_2$ depletion in the membrane; B,C, $Ca^{2+}$-bound desensitized states without $PtdIns(4,5)P_2$; D, apo closed state; E, $Ca^{2+}$-$PtdIns(4,5)$ $P_2$-activated state. Water-soluble short-chain synthetic $PtdIns(4,5)P_2$ diC8 was used in all electrophysiology recordings. **b**, Sample I–V curves corresponding to the various functional states depicted in **a**. **c**, Comparison of relative outward currents of TRPM4 at various functional states shown in **a**. Currents are normalized against the maximum current at the initial $Ca^{2+}$-activated state (state A). Bars represent the mean ± s.d. of $n = 5$ independent replicates (shown as dots). $P$ values were calculated using a two-sided Student's $t$-test. **$P < 0.01$. **d**, Cartoon representation of a single human TRPM4 subunit structure with each domain individually colored. **e**, Overall hTRPM4 structures in different conformational states with the front subunit highlighted. The bound $Ca^{2+}$ (green spheres) and $PtdIns(4,5)P_2$ (gold stick model) are shown in the front subunit.

inositol C4 and C5 phosphate groups, including R664, K925 and K928, markedly mitigate the potentiation effect of $PtdIns(4,5)P_2$ on channel activation (Fig. 2g and Extended Data Fig. 4c). $PtdIns(4,5)P_2$ has been shown to modulate several TRP channels[33]. Intriguingly, $PtdIns(4,5)P_2$ binding in TRPM4 is akin to that in TRPV1 (ref. 34) and TRPV5 (refs. 35–37) but different from that observed in TRPM3 (ref. 38) and TRPM8 (refs. 39–41) (Extended Data Fig. 5).

Two key structural differences between open and closed TRPM4 at the $PtdIns(4,5)P_2$ site imply state-dependent lipid binding with low affinity to the closed channel (Fig. 2h). Firstly, W864 on S3 protrudes

outwardly in the closed state and directly clashes with the glycerol group of $PtdIns(4,5)P_2$. Secondly, Y1057 interacts with the C5 phosphate of $PtdIns(4,5)P_2$ in the open state but rotates toward the position of the inositol group in the closed state and imposes steric hindrance to $PtdIns(4,5)P_2$ binding. Furthermore, no TRPM4 structure with only $PtdIns(4,5)P_2$ bound has been observed. Thus, $PtdIns(4,5)P_2$ binding at its active site likely occurs after initial $Ca^{2+}$ activation, which in turn stabilizes the channel in the open state.

It is worth noting that the $Ca^{2+}$-$PtdIns(4,5)P_2$-bound open TRPM4 structure is virtually identical to the recent structure determined

**Table 1 | Cryo-EM data collection, refinement and validation statistics**

| | Ca²⁺-PtdIns(4,5)P₂ open (EMD-48563), (PDB 9MRT) | Ca²⁺ putative desensitized (EMD-48603), (PDB 9MT8) | Apo closed (EMD-48604), (PDB 9MTA) | ATP-inhibited (EMD-48605), (PDB 9MTC) |
|---|---|---|---|---|
| **Data collection and processing** | | | | |
| Magnification | 165,000 | 165,000 | 165,000 | 105,000 |
| Voltage (kV) | 300 | 300 | 300 | 300 |
| Electron exposure (e⁻ per Å²) | 60 | 60 | 60 | 60 |
| Defocus range (µm) | −0.9 to −2.2 | −0.9 to −2.2 | −0.9 to −2.2 | −0.9 to −2.2 |
| Pixel size (Å) | 0.738 | 0.738 | 0.738 | 0.775 |
| Symmetry imposed | $C_4$ | $C_4$ | $C_4$ | $C_4$ |
| Initial particle images (no.) | 1,491,638 | 1,491,638 | 1,491,638 | 4,103,287 |
| Final particle images (no.) | 32,765 | 57,360 | 78,427 | 153,652 |
| Map resolution (Å) | 2.44 | 3.19 | 2.78 | 2.63 |
| FSC threshold: 0.143 | | | | |
| **Refinement** | | | | |
| Model resolution (Å) FSC threshold: 0.143 | 2.44 | 3.18 | 2.78 | 2.64 |
| Map sharpening B factor (Å²) | −54.8 | −89.6 | −82.0 | −86.3 |
| Model composition | | | | |
| Nonhydrogen atoms | 30,544 | 27,192 | 31,232 | 30,964 |
| Protein residues | 3,804 | 3,420 | 3,924 | 3,876 |
| Ligands | Ca²⁺: 8, PtdIns(4,5)P₂: 4 | Ca²⁺: 4 | - | ATP: 4 |
| B factors (Å²) | | | | |
| Protein | 85.41 | 95.81 | 92.17 | 53.54 |
| Ligand | 57.74 | 43.90 | - | 64.80 |
| Root-mean-square deviations | | | | |
| Bond lengths (Å) | 0.003 | 0.003 | 0.002 | 0.003 |
| Bond angles (°) | 0.510 | 0.450 | 0.435 | 0.497 |
| Validation | | | | |
| MolProbity score | 1.01 | 1.30 | 1.03 | 1.47 |
| Clashscore | 2.33 | 4.09 | 2.45 | 3.86 |
| Poor rotamers (%) | 0.87 | 1.39 | 0.48 | 1.83 |
| Ramachandran plot | | | | |
| Favored (%) | 98.92 | 98.59 | 98.47 | 97.56 |
| Allowed (%) | 1.08 | 1.41 | 1.53 | 2.44 |
| Disallowed (%) | 0 | 0 | 0 | 0 |

at 37 °C in the presence of Ca²⁺ and the positive modulator DVT (Extended Data Fig. 6a). In that study, the high temperature appears to be necessary for DVT to bind at a location near where the PtdIns(4,5)P₂ head group resides and the negatively charged modulator engages in similar electrostatic interactions with several PtdIns(4,5)P₂-interacting residues (Extended Data Fig. 6b). However, our open TRPM4 structure was obtained at 12 °C with native ligands (Methods) and the channel activity was measured using electrophysiological recordings at room temperature, suggesting that higher temperature may facilitate channel opening but is not the determining factor for TRPM4 activation. We suspect that a higher temperature may enhance the dynamic movement of the intracellular domains and facilitate DVT access to its active site. Similar to PtdIns(4,5)P₂, DVT binding stabilizes the channel in the open state.

## Ligand activation mechanism

To understand how calcium and PtdIns(4,5)P₂ activate TRPM4, we compared the structures of TRPM4 in Ca²⁺-PtdIns(4,5)P₂-bound open and apo closed states by aligning them along the pore axis (Fig. 3a).

As TRPM4 activation is initiated by Ca²⁺ binding, the global conformational changes for channel opening have to start from the Ca²⁺ site in S1–S4. Although the initial Ca²⁺-induced local conformational change is subtle, it is progressively propagated to a much larger movement at other parts of the channel through tight intersubunit and intrasubunit packing as sequentially described below and illustrated in Supplementary Video 1.

Firstly, Ca²⁺ binding within the S1–S4 domain drives the N-terminal part of S3 to swing inwardly toward the center of the pocket, resulting in a translation of the W864 side chain toward the C-terminal part of S4. Mediated by its tight packing with H908 on S4, W864 movement pushes the C-terminal part of S4 toward S5 from the neighboring pore domain and initiates a cascade of conformational changes that leads to the opening of the pore and to larger movements at the cytosolic domains as described in the following (Fig. 3b–d and Supplementary Video 2). Substituting W864 or H908, two key residues that couple the movement between S3 and S4, to alanine can mitigate the channel activation (Fig. 3e and Extended Data Fig. 7a).

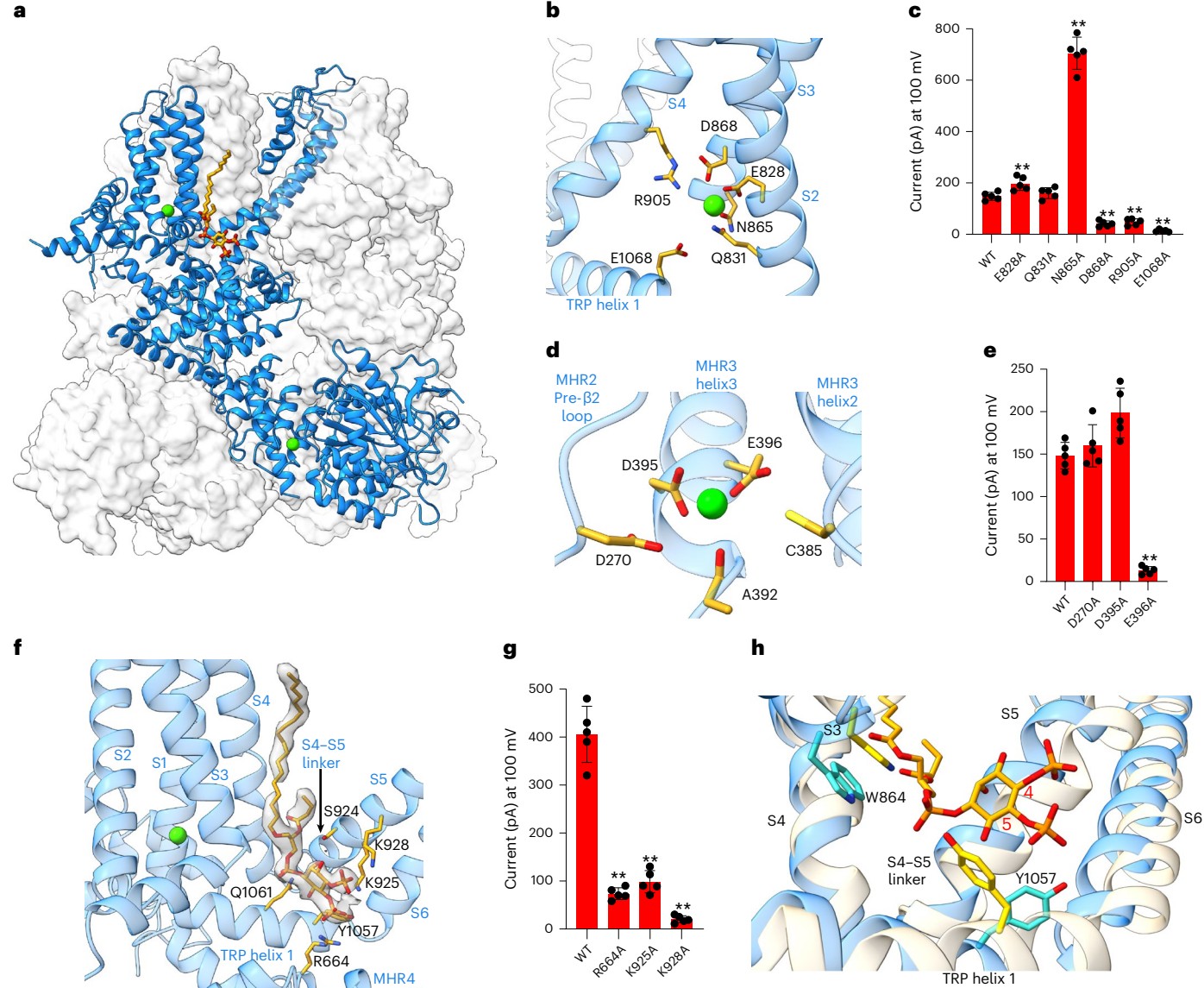

**Fig. 2 | Ca²⁺ and PtdIns(4,5)P₂ binding in hTRPM4. a**, Ca²⁺-PtdIns(4,5)P₂-activated TRPM4. The front subunit, with bound Ca²⁺ (green spheres) and PtdIns(4,5)P₂ (gold sticks), is highlighted in blue cartoon representation. **b**, Zoomed-in view of the transmembrane Ca²⁺-binding site. **c**, Ca²⁺-activated outward currents of wild-type TRPM4 and its transmembrane Ca²⁺-site mutants in the steady state without PtdIns(4,5)P₂. Currents were recorded at +100 mV in inside-out patches with 300 μM Ca²⁺ in the bath (cytosolic). **d**, Zoomed-in view of the intracellular Ca²⁺-binding site. **e**, Ca²⁺-activated outward currents of wild-type TRPM4 and its intracellular Ca²⁺-site mutants in the steady state without PtdIns(4,5)P₂. Currents were recorded at +100 mV in inside-out patches with 300 μM Ca²⁺ in the bath (cytosolic). **f**, Zoomed-in view of the PtdIns(4,5)P₂-binding site. PtdIns(4,5)P₂ and

its interacting residues are shown in stick representation. PtdIns(4,5)P₂ density (gray surface) is contoured at 0.17 in ChimeraX. **g**, PtdIns(4,5)P₂-potentiated outward currents of wild-type TRPM4 and its PtdIns(4,5)P₂-site mutants. Currents were recorded at +100 mV in inside-out patches with 300 μM Ca²⁺ and 10 μM PtdIns(4,5)P₂ diC8 in the bath. **h**, Structural comparison at the PtdIns(4,5)P₂-binding site between the open (blue) and closed (wheat) TRPM4, showing the conformational changes of W864 and Y1057 with their side chains colored in cyan (open state) or yellow (closed state). The red numbers mark the C4 and C5 positions of inositol. For data in **c,e,g**, bars represent the mean ± s.d. of *n* = 5 independent replicates (shown as dots). *P* values were calculated using a two-sided Student's *t*-test. \*\**P* < 0.01.

Secondly, the S4 displacement drives its C-terminal F910 closer to its interacting partner F935 near the cytosolic end of S5 from the neighboring pore domain. In the apo closed state, the F910 side chain makes a direct van der Waal contact with the tip of the F935 aromatic ring. When moving closer upon Ca²⁺ activation, the side chains of F910 and F935 undergo concerted rotation in opposite directions to avoid collision, with the F910 ring rotating inward and the F935 ring rotating outward (Fig. 3b–d and Supplementary Video 2). This rearrangement allows the F910 benzene ring to engage in more extensive hydrophobic contact with the Cα and the main chain of F935, driving an upward swing at the joint region between S5 and the S4–S5 linker (Fig. 3d). Tightly packed with S6, the joint's movement drives the S6

helix to bend away from the central pore axis, resulting in an outward translation of the I1040 and S1044 gating residues and in the opening of the intracellular gate (Fig. 3f,g and Supplementary Video 1). Thus, F910 and F935 have a central role in coupling the Ca²⁺-induced conformational change at S1–S4 to the pore opening in TRPM4. Indeed, their substitutions to alanine lead to a complete loss of channel function (Fig. 3e and Extended Data Fig. 7a). It is worth noting that several phenylalanine residues surrounding F935 also undergo concurrent rotations of their side chain as seen in F935, including F931, F932 and F936 (Extended Data Fig. 7b).

Thirdly, the pore-opening bending movement of S6 is extended to the TRP domain through direct linkage, dragging the TRP helix 1

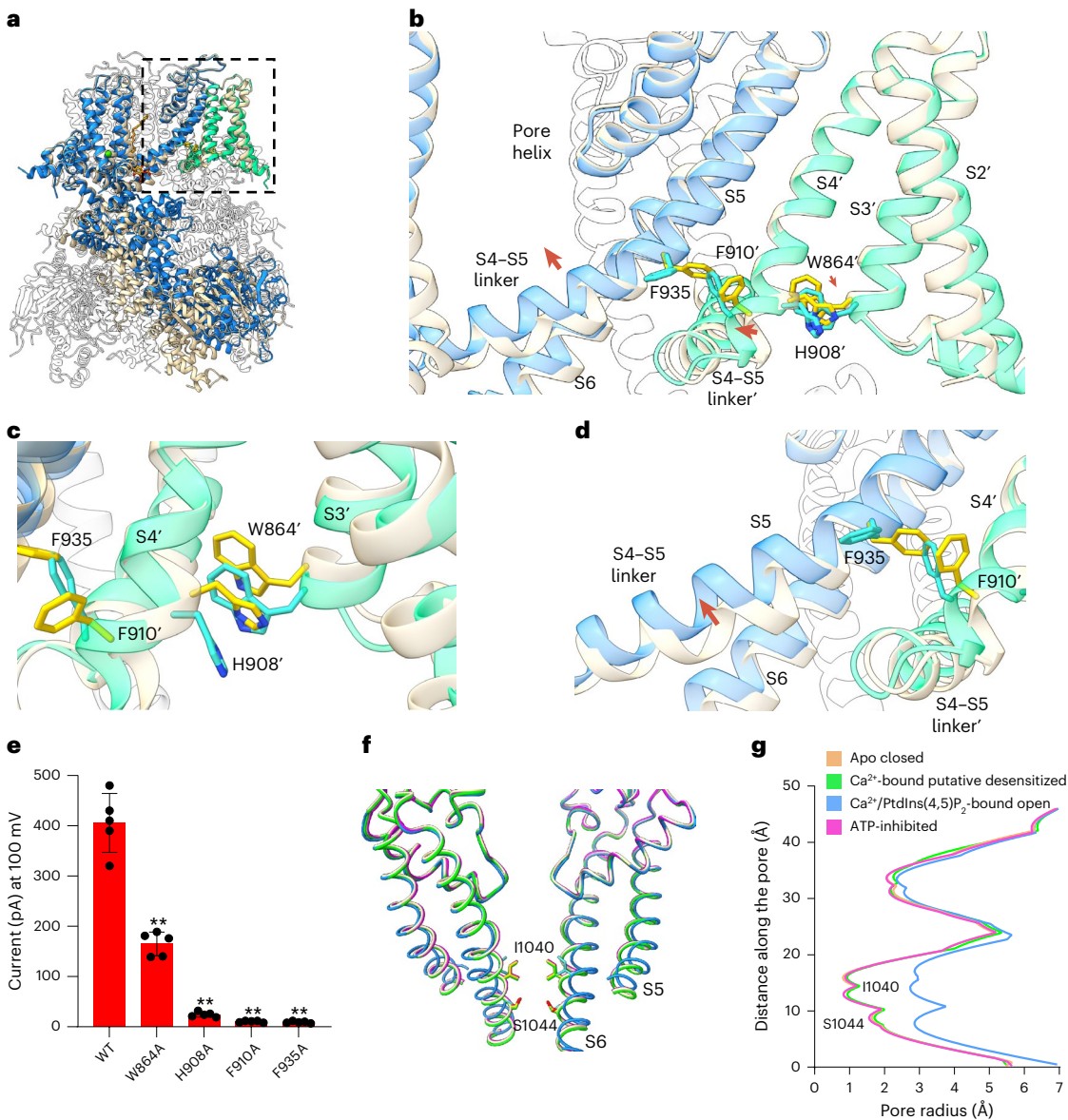

**Fig. 3 | Conformational changes at the transmembrane region upon channel activation. a**, Superposition of the hTRPM4 structures in the open and closed states with the front subunits (open in blue and closed in wheat) and the neighboring S1–S4 domains (open in green and closed in wheat) highlighted in color. **b**, Overview of conformational changes at the pore domain and its neighboring S1–S4 (labeled with single quotation marks) between open and closed TRPM4. Red arrows mark the major movements from closed to open state. Key residues for TRPM4 activation are colored in cyan and yellow for the open and closed states, respectively. The conformational changes at the transmembrane region are visualized in Supplementary Video 2. **c**, Zoomed-in view of the Ca²⁺-induced local conformational change within the S1–S4 domain. **d**, Zoomed-in view of the coupled movement from the S1–S4 domain to the neighboring pore domain upon channel activation. The red arrow marks the upward swing of the joint region between the S4–S5 linker and S5 that leads to the opening of the pore. **e**, Functional effect of substitutions of the residues important for TRPM4 activation. Currents were recorded at +100 mV in inside-out patches with 300 μM Ca²⁺ and 10 μM PtdIns(4,5)P₂ diC8 in the bath. Bars represent the mean ± s.d. of $n = 5$ independent replicates (shown as dots). $P$ values were calculated using a two-sided Student's $t$-test. **$P < 0.01$. **f**, Structural comparison of the TRPM4 ion conduction pore at various states: apo closed (wheat), Ca²⁺-bound putative desensitized (green), Ca²⁺-PtdIns(4,5)P₂-bound open (blue) and ATP-inhibited (pink). Gating residues I1040 and S1044 are show in stick representation. The front and back subunits were removed for clarity. **g**, Pore radius along the central axis in the different TRPM4 states.

into a rotation movement (Fig. 4a,b and Supplementary Video 3). As TRP helix 1 makes extensive interactions with helices 6–7 of the MHR4 domain, its rotation induces a larger rigid-body movement at the cytosolic MHR domains, which swing upward toward the transmembrane domain of the channel (Figs. 4a and 4c and Supplementary Video 3). In addition, MHR1 and MHR2 undergo a rotation movement relative to MHR3 and MHR4 caused by the reorientation of helices 1 and 2 of MHR3 (Supplementary Video 3). Consequently, the MHR domains of TRPM4 engage in completely different sets of intersubunit interactions between open and closed states (Extended Data Fig. 8 and Supplementary Video 4).

As mentioned previously, the movement of W864 and the rotation of Y1057 on TRP helix 1 during channel activation would allow for PtdIns(4,5)P₂ binding, which in turn engages in interactions with multiple domains directly involved in the above-described cascade of conformational changes, including the joint region between the S4–S5 linker and S5 (S924, K925 and K928), TRP helix 1 (Y1057 and Q1061) and the MHR4 domain (R664), thereby stabilizing these regions in their open conformations.

## Putative desensitization mechanism of hTRPM4
The Ca²⁺-bound TRPM4 structure adopts a conformation in between open and closed states; its S1–S4 domain undergoes a similar

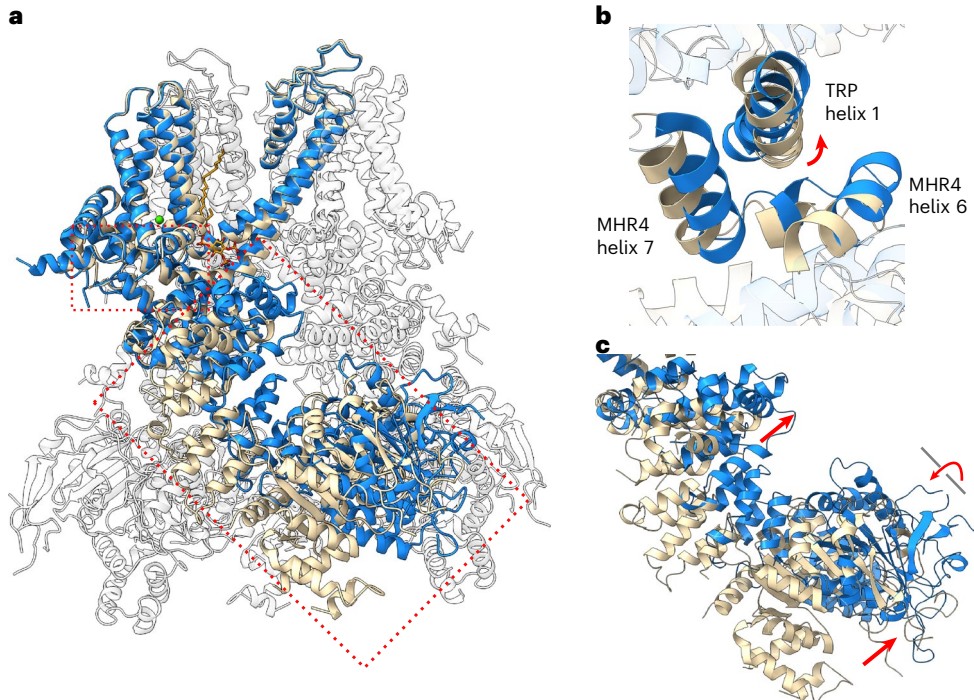

**Fig. 4 | Conformational changes at the cytosolic region of hTRPM4 upon channel activation. a**, Structural comparison of TRPM4 between open and closed states. The front subunits are highlighted in blue (open) and wheat (closed) with TRP domain and MHR domains boxed in red. **b**, Zoomed-in view of the conformational changes at the interface between TRP helix 1 and the MHR4 domain. Red arrow marks the rotation movement of TRP helix 1. **c**, Conformational changes at the MHR domains. Red arrows mark the upward swing of MHRs domains and the rotation of MHR1 and MHR2. The conformational changes at the cytosolic region are visualized in Supplementary Video 3.

$Ca^{2+}$-induced local conformational change as seen in the open TRPM4, whereas the rest of the channel, including the pore and cytosolic domains, remains in the closed conformation (Figs. 3f,g and 5a and Extended Data Fig. 9). Compared to the open and apo closed TRPM4, a key structural change in the $Ca^{2+}$-bound structure occurs at the contact site between F910 and F935, whose interaction is essential for relaying the $Ca^{2+}$-induced local conformational change in S1–S4 to the channel opening (Fig. 3d). In both closed and open structures, the densities for F910 and F935, as well as their surrounding residues, are well defined in the EM maps (Fig. 5b). In the $Ca^{2+}$-bound TRPM4, however, no clear density is observed in the EM map for the benzene ring of F910 and the F935 side chain also becomes poorly defined, indicating the loss of engagement between these two residues (Fig. 5b). As their surrounding residues remain well structured, the loss of density in F910 and F935 is caused by their side-chain mobility rather than the flexibility of the local region. It appears that the F910 side chain flips out from the contact interface between S4 and S5 and becomes flexible. The structure feature of the $Ca^{2+}$-bound TRPM4 points to two possible conformations: a preopen intermediate state in which the $Ca^{2+}$-induced conformational change at S1–S4 has not yet coupled to the pore or a postopen desensitized state in which the channel pore along with its tightly associated cytosolic domain returns to a closed conformation because of the decoupling between F910 and F935. With a close contact between F910 on S4 and F935 on S5 in the apo closed TRPM4, the trajectory of S4 movement upon $Ca^{2+}$ binding in S1–S4 would directly drive the S5 movement for pore opening. Loss of the F910–F935 contact because of their side-chain movement in the $Ca^{2+}$-bound structure likely occurs after pore opening. This structural change decouples the driving force between $Ca^{2+}$-induced S4 movement and pore opening, allowing the pore domain and its associated TRP and cytosolic domains to return to the closed conformation (Supplementary Video 5). We, therefore, hypothesize that the $Ca^{2+}$-bound TRPM4 structure represents a putative desensitized state. Thus, the $Ca^{2+}$-activated channel is transiently stable and requires $PtdIns(4,5)P_2$ binding to retain channel conduction. Upon the loss of $PtdIns(4,5)P_2$ stabilization, the pore domain is prone to return to its closed state, resulting in the side-chain slip between F910 and F935 that decouples the conformational changes between S4 and S5 and renders the channel desensitized to $Ca^{2+}$ activation (Supplementary Video 5).

## ATP inhibition of TRPM4

Mg-free ATP is known to inhibit TRPM4 by binding to the N-terminal MHR domains[25]. To recapitulate the effect of ATP on $Ca^{2+}$-$PtdIns(4,5)P_2$-activated TRPM4, we prepared the protein sample in the presence of $Ca^{2+}$, $PtdIns(4,5)P_2$ and $Mg^{2+}$-free ATP (in $Na^+$ salt) and obtained a 2.7 Å structure in the closed conformation (Figs. 3f,g and 6, Extended Data Fig. 10a and Table 1). Despite the presence of $PtdIns(4,5)P_2$ and $Ca^{2+}$, we only observed bound ATP but not $Ca^{2+}$ or $PtdIns(4,5)P_2$ in the structure, indicating that ATP abrogates $PtdIns(4,5)P_2$ and $Ca^{2+}$ binding by stabilizing the channel in the closed conformation. ATP binds at the same intersubunit interface between MHR1 and MHR3 as observed in the apo closed TRPM4 structure (Fig. 6 and Extended Data Fig. 10b,c). In the apo structure, the side-chain density of those interfacial residues is poorly defined in the EM map, suggesting weak intersubunit contact at this interface (Extended Data Fig. 10d). The bound ATP stabilizes these interfacial residues from both subunits by engaging in extensive interactions with them (Fig. 6b and Extended Data Fig. 10e). Substitutions of several ATP-interacting residues were previously shown to mitigate ATP inhibition[25]. It is worth noting that there is no observable density for the γ-phosphate of ATP in the EM map (Extended Data Fig. 10c), suggesting that this phosphate group is flexible and does not interact with the protein. This is consistent with the functional observation that adenosine diphosphate and ATP have equivalent inhibitory effects on TRPM4 (ref. 23). Thus, free ATP functions like a molecular glue that solidifies the intersubunit contact between MHR1 and MHR3. As the cytosolic MHR domains have to undergo concerted movement with

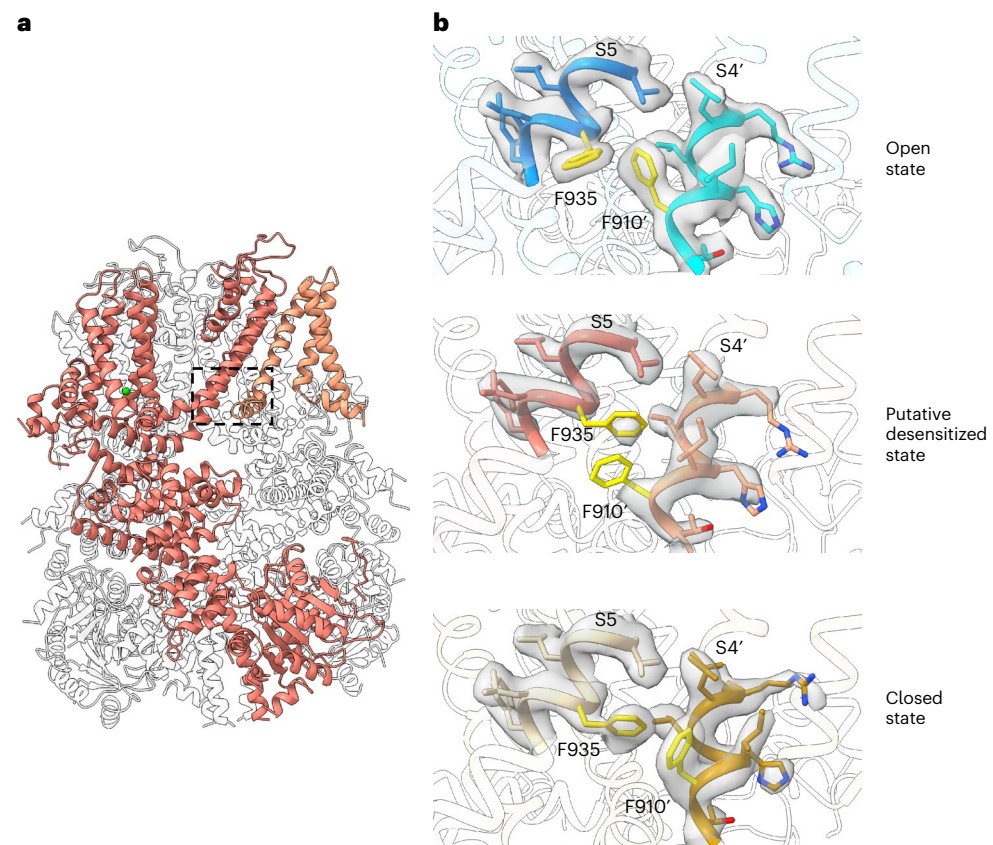

**Fig. 5 | Putative desensitization of hTRPM4. a**, Structure of Ca²⁺-bound putative desensitized TRPM4 with the F935–F910′ interface region boxed. The front subunit and its neighboring S1–S4 domain are highlighted in salmon and light salmon, respectively. **b**, Zoomed-in views of the F935–F910′ interface regions in the open (top), putative desensitized (middle) and closed (bottom) states. The F910′ and F935 side chains are shown in yellow. The EM density maps for F935, F910′ and their surrounding areas are shown in gray surface contoured at 0.2 in ChimeraX.

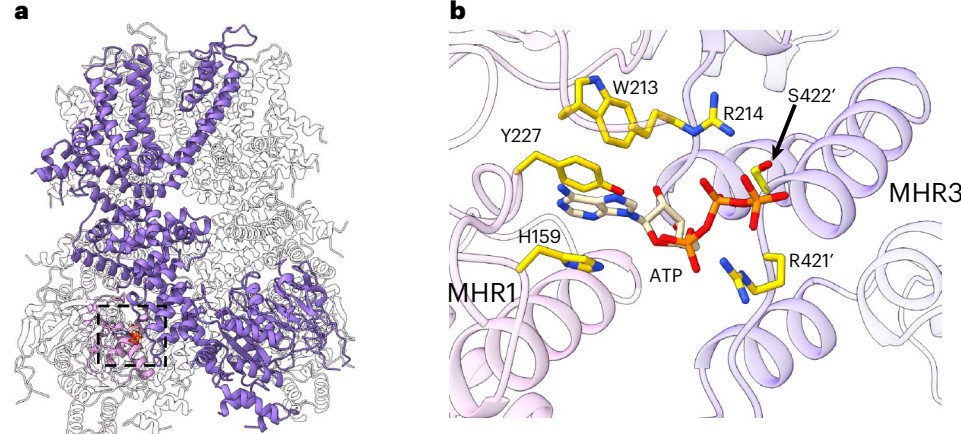

**Fig. 6 | ATP inhibition of hTRPM4. a**, Structure of ATP-bound inhibited TRPM4 with one ATP-binding site boxed. The front subunit is highlighted in purple and the region in the neighboring subunit that participates in ATP binding is highlighted in pink. **b**, Zoomed-in view of the ATP-binding site with the side chains of ATP-interacting residues shown as yellow sticks.

the transmembrane domain upon channel activation, the ATP binding would lock these cytosolic domains in the closed conformation and prevent them from undergoing any conformational change in response to Ca²⁺-PtdIns(4,5)P₂ activation.

## Discussion

In this study, we determined human TRPM4 structures in various conformational states, allowing us to elucidate the structural mechanisms of TRPM4 regulation by the key signaling molecules Ca²⁺, PtdIns(4,5)P₂

and ATP (Supplementary Videos 1–5). Through a structural comparison, it becomes clear that channel opening entails a series of concerted movements that start at the transmembrane domain and propagate to the cytosolic domains. Within the transmembrane domain, calcium binding triggers the initial conformational changes in the S1–S4 domain, which then couple to movement at the joint between the S4–S5 linker and S5 in the neighboring pore domain and ultimately lead to the pore opening through the outward bending of the pore-lining S6 helix. Through direct linkage, the pore opening S6 movement drives

a rotation of the TRP helix 1, which, mediated by the tight interdomain interactions, leads to an upward swing of the cytosolic MHR domains. Consequently, the cytosolic MHR domains engage in different sets of intersubunit contacts between open and closed states.

While TRPM4 activation is initiated by cytosolic $Ca^{2+}$ binding in S1–S4, the $Ca^{2+}$-activated TRPM4 appears to be transiently stable and requires the binding of $PtdIns(4,5)P_2$ to maintain its conductivity. However, the channel needs to be in an open conformation before $PtdIns(4,5)P_2$ can access its active site. In other words, $Ca^{2+}$-initiated activation is a prerequisite for $PtdIns(4,5)P_2$ potentiation of TRPM4. Once bound, $PtdIns(4,5)P_2$ mediates interactions with multiple domains that help stabilize the channel in the open conformation. Without $PtdIns(4,5)P_2$ stabilization, the side-chain rotation of the two central phenylalanine residues (F910 and F935) at the interface between S4 and S5 of the neighboring subunit can disengage the $Ca^{2+}$-induced conformational change at S1–S4 from driving the pore opening, causing TRPM4 desensitization despite the presence of $Ca^{2+}$. The concerted long-range movements from the pore to the cytosolic domains between open and closed states underlie the allosteric inhibition of TRPM4 by ATP. By binding at an intersubunit interface formed at the closed state between MHR1 and MHR3 of two neighboring subunits, ATP fastens this intersubunit contact and locks the cytosolic domains in a closed conformation, thereby inactivating the channel by preventing the concerted activation movement from the distal end.

$PtdIns(4,5)P_2$ also acts as a positive modulator for the temperature-sensitive TRPM3 and TRPM8 channels[33]. Structural studies of $PtdIns(4,5)P_2$-bound TRPM3 and TRPM8—using the short-chain lipid $PtdIns(4,5)P_2$ diC8—revealed a binding site distinct from that of TRPM4 (Extended Data Fig. 5a–c). In both TRPM3 (ref. [38]) and TRPM8 (refs. [39–41]), $PtdIns(4,5)P_2$ occupies a pocket formed by the pre-S1 domain, S1 and S4 helices, S4–S5 linker and TRP helix 1. Notably, most of these structures depict closed channel conformations, even in the presence of agonists. An open-state TRPM8 structure was only observed when $PtdIns(4,5)P_2$ was coapplied with two agonists, C3 and AITC (Extended Data Fig. 5c).

The TRPV1 and TRPV5 channels are modulated by $PtdIns(4,5)P_2$ as well, although its effects on TRPV1 are complex, with evidence for both positive and negative modulation[33]. Several lipid-bound TRPV1 structures were determined in a recent study using long-chain brominated $PtdIns(4,5)P_2$ di(18:1) or short-chain $PtdIns(4,5)P_2$ diC8 (ref. [34]). Structures in the closed channel state were determined with both lipids bound at the vanilloid-binding pocket (Extended Data Fig. 5d), in a similar configuration to that observed in TRPM4. Interestingly, a dilated channel state was determined in the presence of short-chain $PtdIns(4,5)P_2$ diC8 (Extended Data Fig. 5e), where the lipid sits at a higher position in the vanilloid pocket. These observations led to the proposition that $PtdIns(4,5)P_2$ exerts different modulatory effects on TRPV1 according to its binding mode, at least for the short-chain lipid form. $PtdIns(4,5)P_2$ is required for the constitutive activation of the $Ca^{2+}$-selective TRPV5 channel. Several lipid-bound TRPV5 structures were determined in an open configuration using $PtdIns(4,5)P_2$ diC8 (refs. [35–37]), with $PtdIns(4,5)P_2$ being observed in a pocket near the vanilloid-binding site (Extended Data Fig. 5f).

## Online content

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

## Methods

### Protein expression and purification

Full-length human TRPM4 (UniProtKB Q8TD43) containing an N-terminal FLAG-tag (DYKDDDDK) was cloned into a pEZT-BM plasmid[42]. *Escherichia coli* DH10Bac cells were used to synthesize the bacmid that was applied in baculovirus production in Sf9 cells (Thermo Fisher Scientific, 11496015) using Cellfectin II reagent (Thermo Fisher Scientific). For protein expression, HEK293S GnTI⁻ cells (American Type Culture Collection (ATCC), CRL-3022) grown in suspension to a density of $3 \times 10^6$ cells per ml were infected with P3 viruses at a ratio of 1:40 (virus to cell, v/v) and supplemented with 10 mM sodium butyrate to boost protein expression. The cells were then incubated at 37 °C for 48 h before being harvested by centrifugation (5,000$g$, 15 min, 4 °C). The cell pellet was resuspended in lysis buffer (50 mM Tris-HCl pH 7.5 and 150 mM NaCl) supplemented with protease inhibitors (0.5 µg ml⁻¹ pepstatin, 2 µg ml⁻¹ leupeptin, 1 µg ml⁻¹ aprotinin and 1 mM PMSF) and homogenized by sonication. TRPM4 was then extracted with 1% (w/v) lauryl maltose neopentyl glycol (LMNG; Anatrace) in gentle agitation for 2 h at 4 °C. The supernatant was subsequently collected by centrifugation (40,000$g$, 40 min, 4 °C) and incubated with anti-DYKDDDDK G1 affinity resin (GenScript) for 1 h and 30 min at 4 °C in gentle agitation. The resin was then washed with buffer A (50 mM Tris-HCl pH 7.5, 150 mM NaCl and 0.01% LMNG) and elution was performed by incubating for 45 min at room temperature in gentle agitation with buffer B (50 mM Tris-HCl pH 7.5, 150 mM NaCl, 0.01% LMNG, 0.2 mg ml⁻¹ FLAG peptide and 70 µM brain PtdIns(4,5)P₂ (porcine, ammonium salt; Avanti)). The eluate was then concentrated and further purified by size-exclusion chromatography in buffer C (50 mM Tris-HCl pH 7.5, 150 mM NaCl and 0.0035% LMNG) on a Superose 6 Increase 10/300 GL (GE Healthcare).

### Cryo-EM sample preparation and data acquisition

hTRPM4 was concentrated to ~8 mg ml⁻¹ and supplemented with 0.2 mM $CaCl_2$ and 0.1 mM brain PtdIns(4,5)P₂. When needed, 1 mM ATP-Na was also added. Supplemented samples were incubated with the respective ligands for 1 h on ice before vitrification. A 4-µl sample was then applied to a glow-discharged Quantifoil R1.2/1.3 300-mesh gold holey carbon grid (Quantifoil, Micro Tools), blotted for 3.5 s with a blot force of 15 under 100% humidity at 12 °C and plunged into liquid ethane using a Mark IV Vitrobot (FEI).

For the dataset of the sample containing calcium and PtdIns(4,5)P₂, micrographs were acquired on a Titan Krios microscope (FEI) operated at 300 kV with a Falcon4 electron detector (Thermo Fisher), using a slit width of 20 eV on a postcolumn Selectris X energy filter (Thermo Fisher). Data were collected with SerialEM using a Falcon4 camera with a pixel size of 0.738 Å. The defocus range was set from −0.9 to −2.2 µm. Each video was dose-fractionated to 60 frames with a dose rate of 1 e⁻ per Å² per frame for a total dose of 60 e⁻ per Å². The total exposure time was between 3.5 to 4 s.

For the dataset of the sample containing calcium, PtdIns(4,5)P₂ and ATP, micrographs were acquired on a Titan Krios microscope (FEI) operated at 300 kV with a K3 Summit direct electron detector (Gatan), using a slit width of 20 eV on a GIF-Quantum energy filter. Data were collected with SerialEM using the correlated double sampling mode of the K3 camera with a super-resolution pixel size of 0.4135 Å. The defocus range was set from −0.9 to −2.2 µm. Each video was dose-fractionated to 60 frames with a dose rate of 1 e⁻ per Å² per frame for a total dose of 60 e⁻ per Å². The total exposure time was between 5 and 6 s.

### Cryo-EM data processing

The workflow for data processing is summarized in Extended Data Figs. 2 and 10a. Data processing was performed using cryoSPARC[43] following the general scheme described below with some modifications between the datasets.

Videos were subjected to patch motion correction and subsequent patch contrast transfer function (CTF) estimation. The resulting micrographs were curated to remove images with bad defocus values, ice contamination and carbon. An initial round of particle picking was carried out with blob picker. Particles were then extracted and subjected to one round of two-dimensional (2D) classification. Classes displaying clear features of the TRPM4 channel were selected and used to repick particles with template picker. Additional rounds of 2D classification were further performed and particles from selected classes were used to obtain an initial three-dimensional (3D) ab initio reconstruction. Several rounds of 3D heterogeneous refinement were then used to remove junk particles. The resulting particles were subjected to 3D classification without alignment to differentiate channel conformations. The best-resolved 3D classes were reextracted with the original pixel size and refined using nonuniform refinement with imposed $C_4$ symmetry[44]. During the refinement, defocus refinement with optimized per-particle defocus and global CTF refinement with optimization of per-group CTF parameters were enabled. Map resolutions were reported according to the gold-standard Fourier shell correlation (FSC) using the 0.143 criterion[45]. Local resolutions and angular distributions were estimated in cryoSPARC.

### Model building

Initial models were obtained using a combination of previously available hTRPM4 structures (Protein Data Bank (PDB) 6BQV and PDB 6BQR) and ModelAngelo[46]. The models were then manually adjusted in Coot[47] and refined against the respective maps in PHENIX[48]. Ligand restraints CIF files were generated using the Grade2 Web Server (Global Phasing). The geometry statistics of the models were obtained using MolProbity[49]. All structural figures were prepared using UCSF ChimeraX[50,51]. Pore radii were calculated using the HOLE program[52].

### Electrophysiology

*TRPM4* WT and respective mutants were cloned into a pEGFP-N1 plasmid (Clontech). First, 1 µg of plasmid was transfected into HEK293 cells (ATCC, CRL-1573) using Lipofectamine 2000 (Life Technology). Then, 48 h after transfection, cells were dissociated by trypsin treatment and kept in a complete serum-containing medium before being replated onto 35-mm tissue culture dishes and incubated in a tissue culture incubator until recording. Mutant constructs for electrophysiological recordings were generated by site-directed mutagenesis using the QuikChange method and verified by sequencing.

Channel currents were recorded in excised inside-out patches. The long-chain native brain PtdIns(4,5)P₂ is insoluble in water and forms liposomes in the recording solutions, making it difficult to fuse into the patch membrane. Therefore, the water-soluble short-chain synthetic PtdIns(4,5)P₂ diC8 (Echelon Bioscience) was used to measure the PtdIns(4,5)P₂ modulation of TRPM4 in most experiments except the one shown in Extended Data Fig. 1, in which the native porcine brain PtdIns(4,5)P₂ was used. The standard bath solution (cytosolic side) contained 145 mM cesium methanesulfonate, 5 mM NaCl, 1 mM $MgCl_2$, 0.3 mM $CaCl_2$ and 10 mM HEPES buffered with Tris, pH 7.4. For the calcium-free condition, 0.5 mM EGTA was added to the bath solution without $CaCl_2$. When required, 10 µM PtdIns(4,5)P₂ diC8 was added to the bath solution. The pipette solution (extracellular side) contained 140 mM sodium methanesulfonate, 1 mM $MgCl_2$, 5 mM $CaCl_2$ and 10 mM HEPES buffered with Tris, pH 7.4. Patch pipettes were pulled from borosilicate glass (Harvard Apparatus) and heat-polished to a resistance of 3–5 MΩ. After the patch pipette was attached to the cell membrane, a giga seal (>10 GΩ) was formed by gentle suction. The inside-out configuration was formed by pulling the pipette away from the cell and the pipette tip was exposed to air for a short time in some cases. The holding potential was set to 0 mV. The current and voltage relationship (I–V curve) was obtained using voltage pulses ramp from −100 to +100 mV over an 800-ms duration. Data were acquired using an AxoPatch 200B amplifier (Molecular Devices) and a low-pass analog filter set to 1 kHz. The current signal was sampled at a rate of 20 kHz

using a Digidata 1322A digitizer (Molecular Devices) and further analyzed with pClamp 11 software (Molecular Devices) and Origin2021b (OriginLab). Sample traces for the I–V curves of macroscopic currents shown were obtained from recordings on the same patch. All data points are the mean ± s.d. of five measurements from different patches ($n$ = 5 independent replicates).

### Statistics and reproducibility

Statistical analysis was performed using Origin2021b. Statistical significance was calculated using a two-sided Student's $t$-test, with significance assumed if $P < 0.01$. In all figure legends, $n$ represents the number of independent replicates. All quantitative data were presented as the mean ± s.d. Exact $P$ values are provided in the respective Source Data files.

### Reporting summary

Further information on research design is available in the Nature Portfolio Reporting Summary linked to this article.

### Data availability

The cryo-EM density maps of the human TRPM4 were deposited to the EM Data Bank under accession numbers EMD-48563 (Ca$^{2+}$-PtdIns(4,5)P$_2$ open), EMD-48603 (Ca$^{2+}$ putative desensitized), EMD-48604 (apo closed) and EMD-48605 (ATP-inhibited). Atomic coordinates were deposited to the PDB under accession numbers 9MRT (Ca$^{2+}$-PtdIns(4,5)P$_2$ open), 9MT8 (Ca$^{2+}$ putative desensitized), 9MTA (apo closed) and 9MTC (ATP-inhibited). All other data and materials supporting the findings of this study can be obtained from the corresponding author upon reasonable request. Source data are provided with this paper.

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

### Acknowledgements

Single-particle cryo-EM data were collected at the University of Texas Southwestern Medical Center Cryo-EM Facility funded by the Cancer Prevention and Research Institute of Texas (CPRIT) Core Facility Support Award RP170644 and at Howard Hughes Medical Institute Janelia Cryo-EM Facility. Cryo-EM sample grids were prepared at the Structural Biology Laboratory at UT Southwestern Medical Center partially supported by grant RP170644 from CPRIT. This work was supported in part by the Howard Hughes Medical Institute (to Y.J.) and by grants from the National Institute of Health (R35GM140892 to Y.J.) and the Welch Foundation (grant I-1578 to Y.J.).

### Author contributions

C.M.T-D. prepared the samples and performed the data acquisition, image processing and structure determination. W.Z. performed the electrophysiology recording. Y.J. supervised the work. All authors participated in research design, data analysis, discussion and manuscript preparation.

### Competing interests

The authors declare no competing interests.

### Additional information

**Extended data** is available for this paper at https://doi.org/10.1038/s41594-025-01705-3.

**Correspondence and requests for materials** should be addressed to Youxing Jiang.

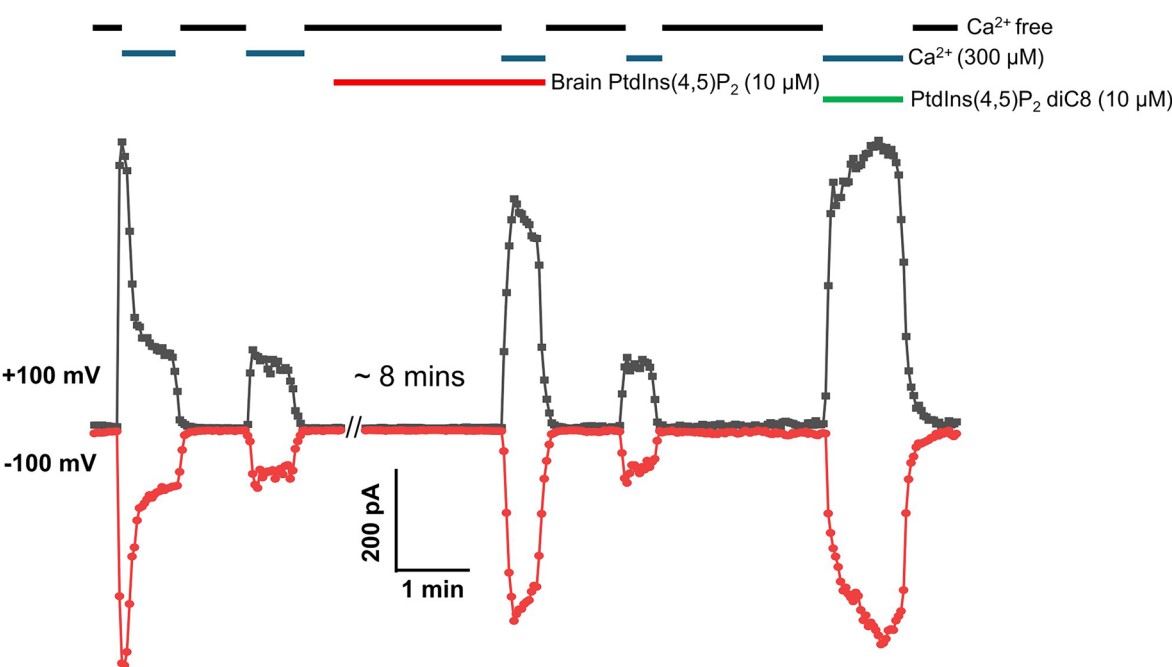

**Extended Data Fig. 1 | Comparison of hTRPM4 potentiation by the long-chain native brain PtdIns(4,5)P$_2$ and short-chain PtdIns(4,5)P$_2$ diC8.** The sample trace represents macroscopic currents of TRPM4 overexpressed in HEK293 cells at ±100 mV in an inside-out patch with the presence or absence of Ca$^{2+}$ and long- or short-chain PtdIns(4,5)P$_2$ in the bath (cytosolic). The water-insoluble long-chain native PtdIns(4,5)P$_2$ was prepared as a 5 mM liposome stock by resuspension and sonication in water. Before recording, the liposome-containing bath solution at 10 μM was thoroughly sonicated to facilitate the lipid vesicle fusion into the patch membrane. A long perfusion time (~ 8 min in this experiment) is needed for sufficient lipid vesicles to fuse into the membrane and potentiate TRPM4 activity in the presence of Ca$^{2+}$. In addition, Ca$^{2+}$-activated phospholipase C can quickly reduce the PtdIns(4,5)P$_2$ level in the membrane, which decreases the potentiation effect of the native PtdIns(4,5)P$_2$. The water-soluble short-chain PtdIns(4,5)P$_2$ diC8, on the contrary, can rapidly fuse into the membrane and effectively potentiate TRPM4 activity.

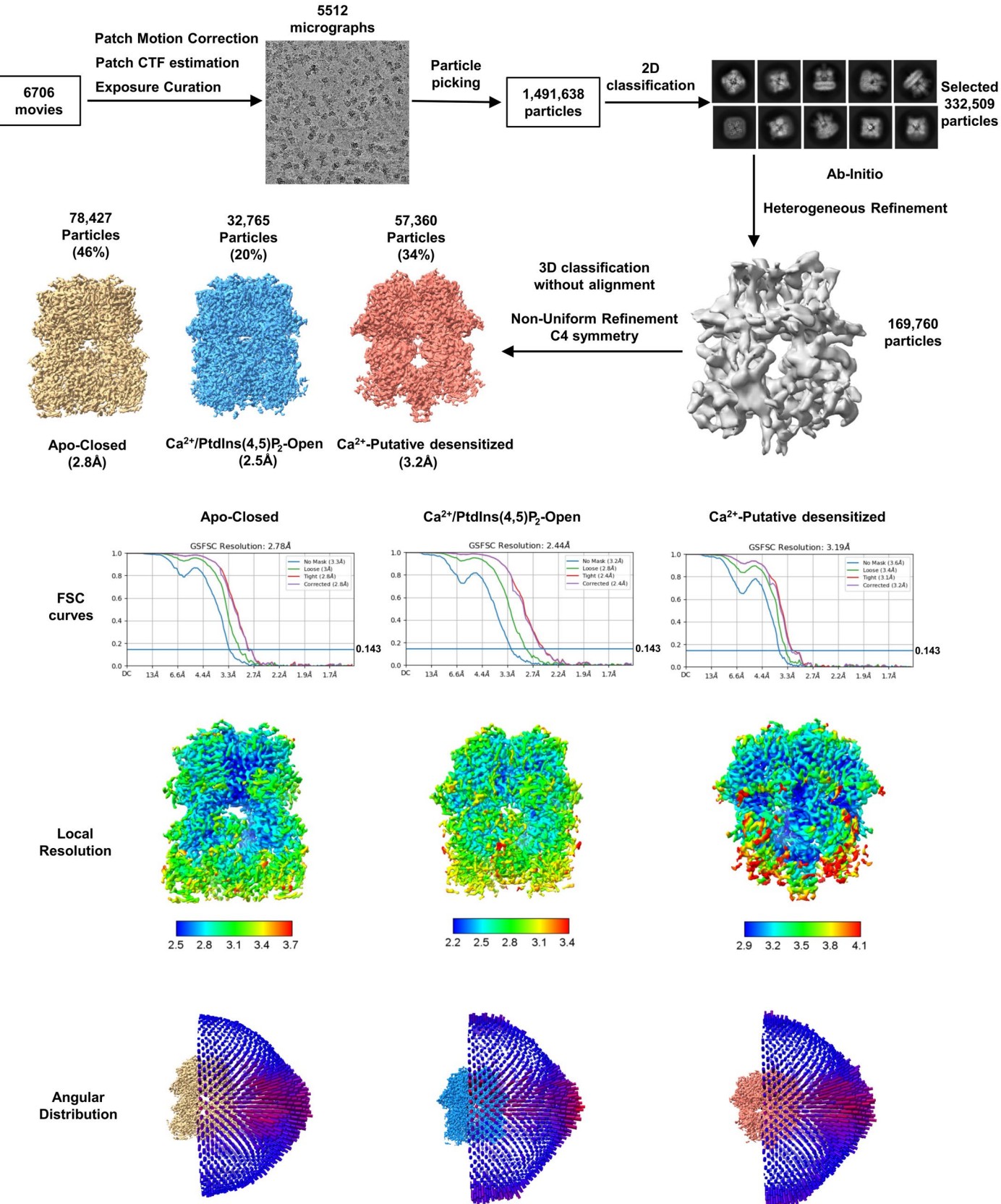

**Extended Data Fig. 2 | Cryo-EM data processing scheme for the hTRPM4 sample prepared in the presence of Ca²⁺ and PtdIns(4,5)P₂.** The single particles used for final structural determination were partitioned into three classes, yielding hTRPM4 structures in apo closed, Ca²⁺-PtdIns(4,5)P₂-bound open, and Ca²⁺-bound putative desensitized states.

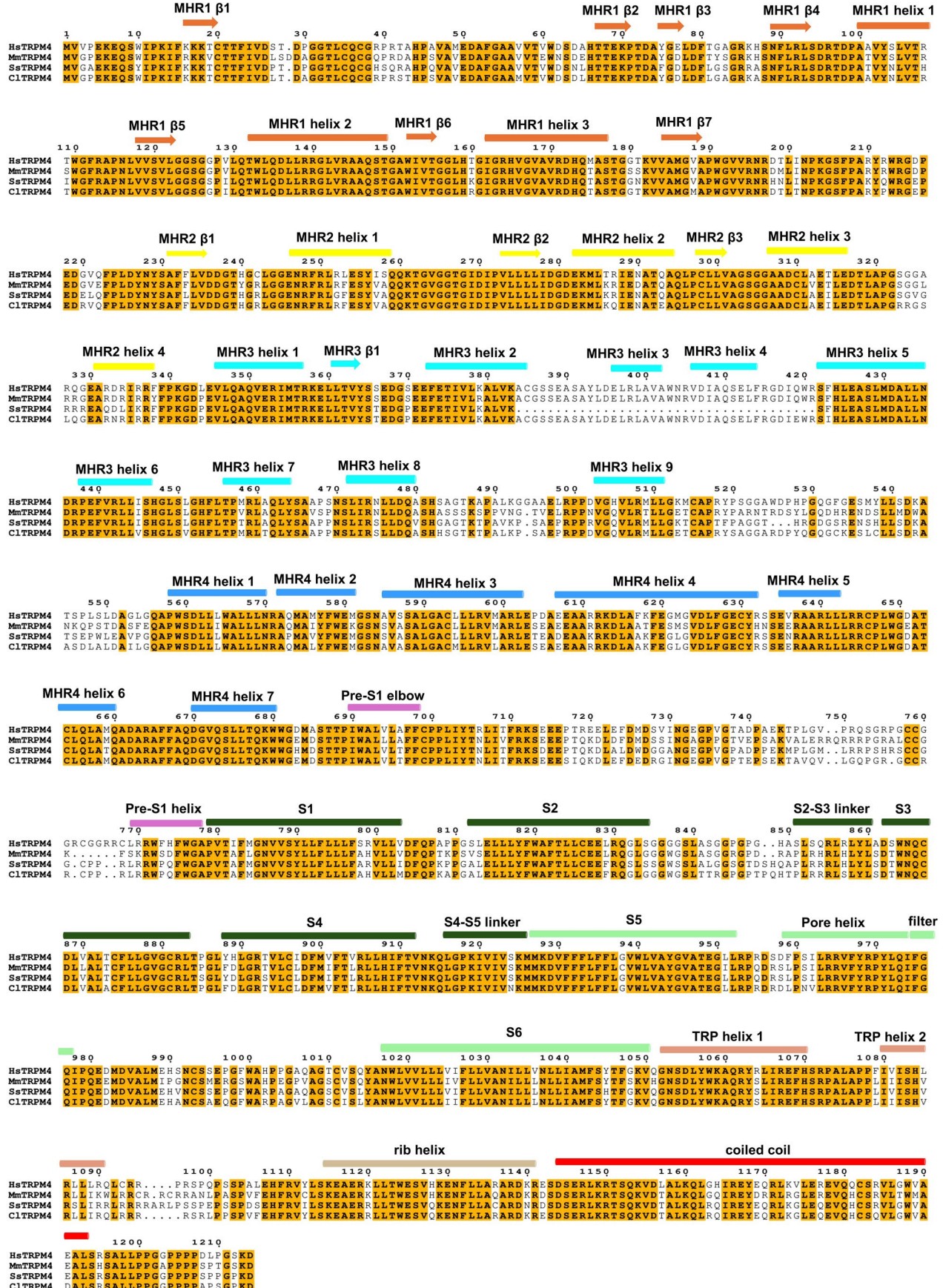

**Extended Data Fig. 3 | Sequence alignment of mammalian TRPM4 channels from human (HsTRPM4), mouse (MmTRPM4), pig (SsTRPM4), and dog (ClTRPM4).** Secondary structure assignment is based on the human TRPM4 structure.

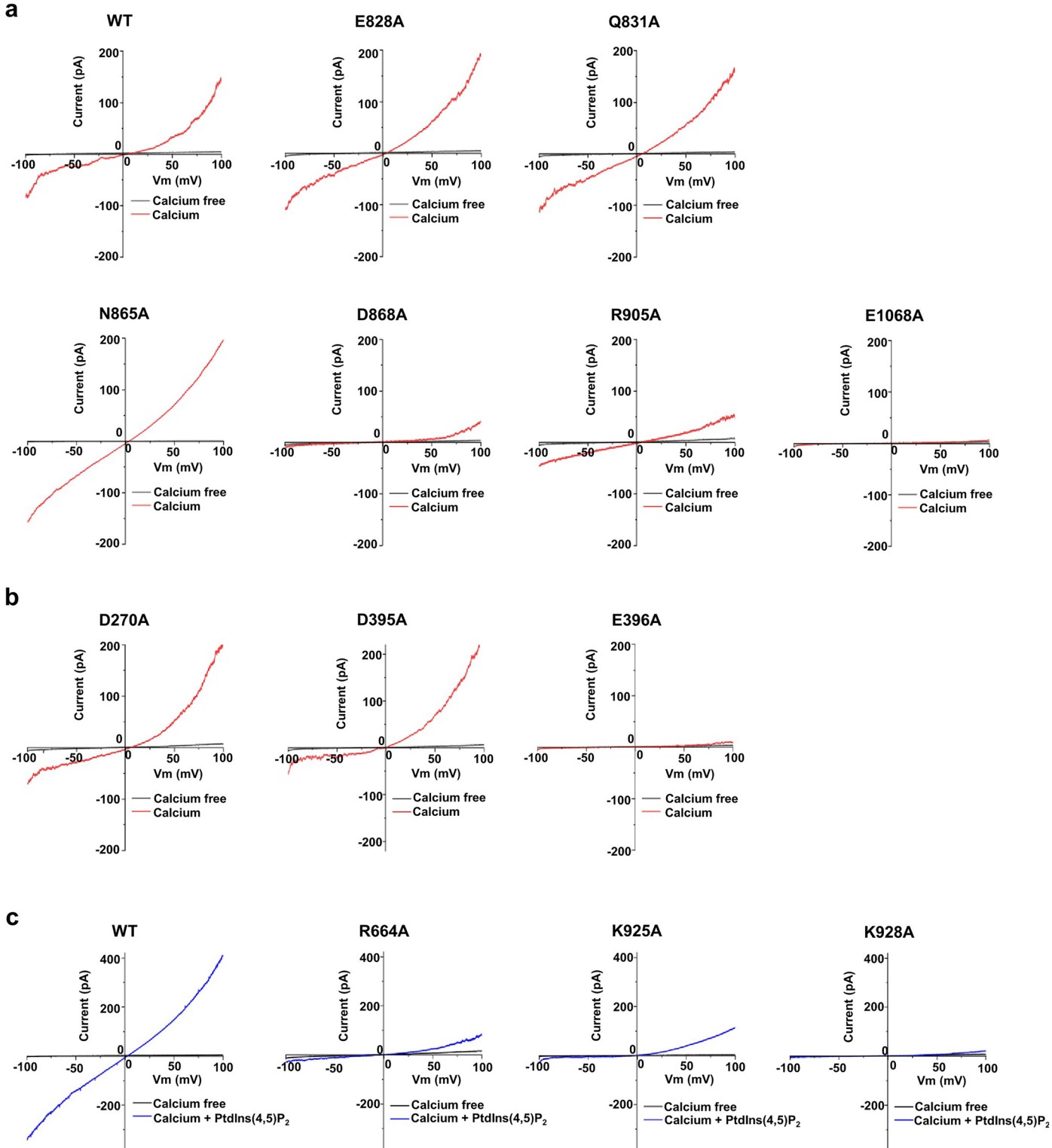

**Extended Data Fig. 4 | Sample I-V curves of hTRPM4 and its mutants at the ligand-binding sites. a** I-V curves of wild-type hTRPM4 and its mutants at the transmembrane $Ca^{2+}$-binding site recorded at steady state in inside-out patches with 300 μM $Ca^{2+}$ in the bath (cytosolic). Currents at +100 mV were used to plot Fig. 2c. **b** I-V curves of hTRPM4 mutants at the intracellular $Ca^{2+}$-binding site recorded at steady state in inside-out patches with 300 μM $Ca^{2+}$ in the bath (cytosolic). Currents at +100 mV were used to plot Fig. 2e. **c** I-V curves of wild-type hTRPM4 and its mutants at the PtdIns(4,5)P$_2$-binding site recorded in inside-out patches with 300 μM $Ca^{2+}$ and 10 μM PtdIns(4,5)P$_2$ di-C8 in the bath (cytosolic). Currents at +100 mV were used to plot Fig. 2g.

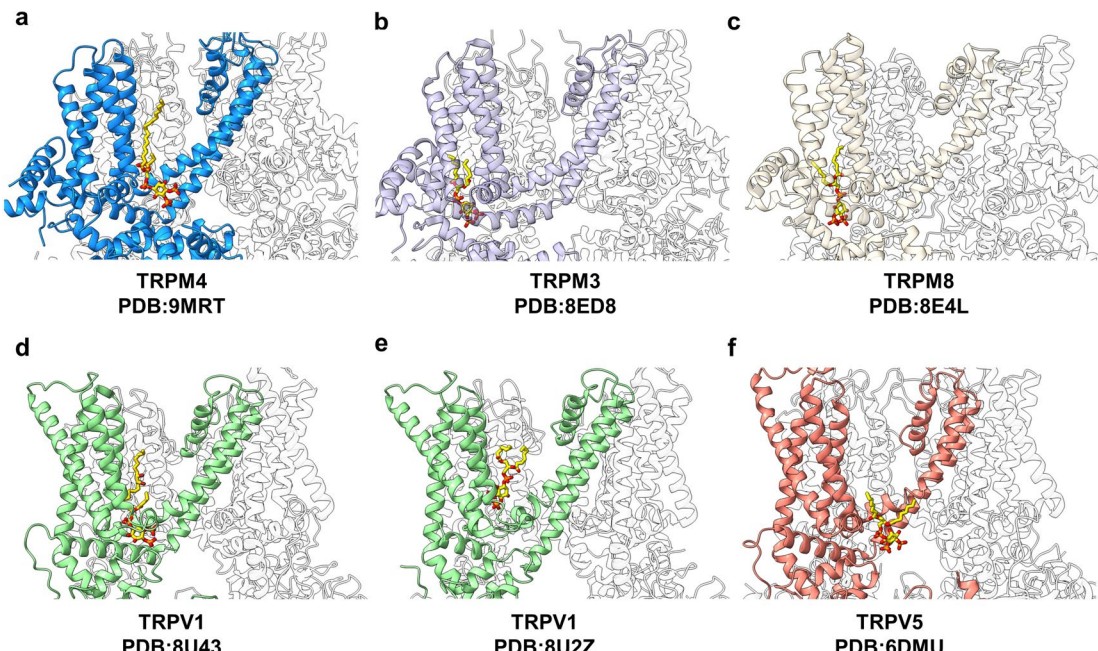

**Extended Data Fig. 5 | Comparison of PtdIns(4,5)P$_2$ binding in the transmembrane regions of various TRP channels.** Only the front subunits with bound PtdIns(4,5)P$_2$ (yellow sticks) are highlighted in color. **a** *Homo sapiens* TRPM4 structure in an open state obtained in the presence of Ca$^{2+}$ and native brain PtdIns(4,5)P$_2$ (this study); **b**, *Mus musculus* TRPM3 structure in a closed state obtained in the presence of short-chain PtdIns(4,5)P$_2$ diC8 and chemical agonist PregS[38]. PregS was not observed in the structure. **c**, *Mus musculus* TRPM8 structure in an open state obtained in the presence of Ca$^{2+}$, short-chain PtdIns(4,5)P$_2$ diC8 and two agonists, cryosim-3 (C3) and allyl isothiocyanate (AITC)[40]. Ca$^{2+}$ and the two agonists are not shown here for simplicity. **d**, *Rattus norvegicus* TRPV1 structure in a closed state obtained in the presence of long-chain brominated PtdIns(4,5)P$_2$ di(18:1)[34]. **e**, *Rattus norvegicus* TRPV1 structure in a dilated state obtained in the presence of short-chain PtdIns(4,5)P$_2$ diC8[34]. In the same study, a closed TRPV1 structure in complex with PtdIns(4,5)P$_2$ diC8 was also observed (PDB: 8U30), in which the short-chain lipid binds at a lower position similar to where long-chain brominated PtdIns(4,5)P$_2$ di(18:1) binds shown in **d**. **f**, *Oryctolagus cuniculus* TRPV5 structure in an open state obtained in the presence of short-chain PtdIns(4,5)P$_2$ diC8[35].

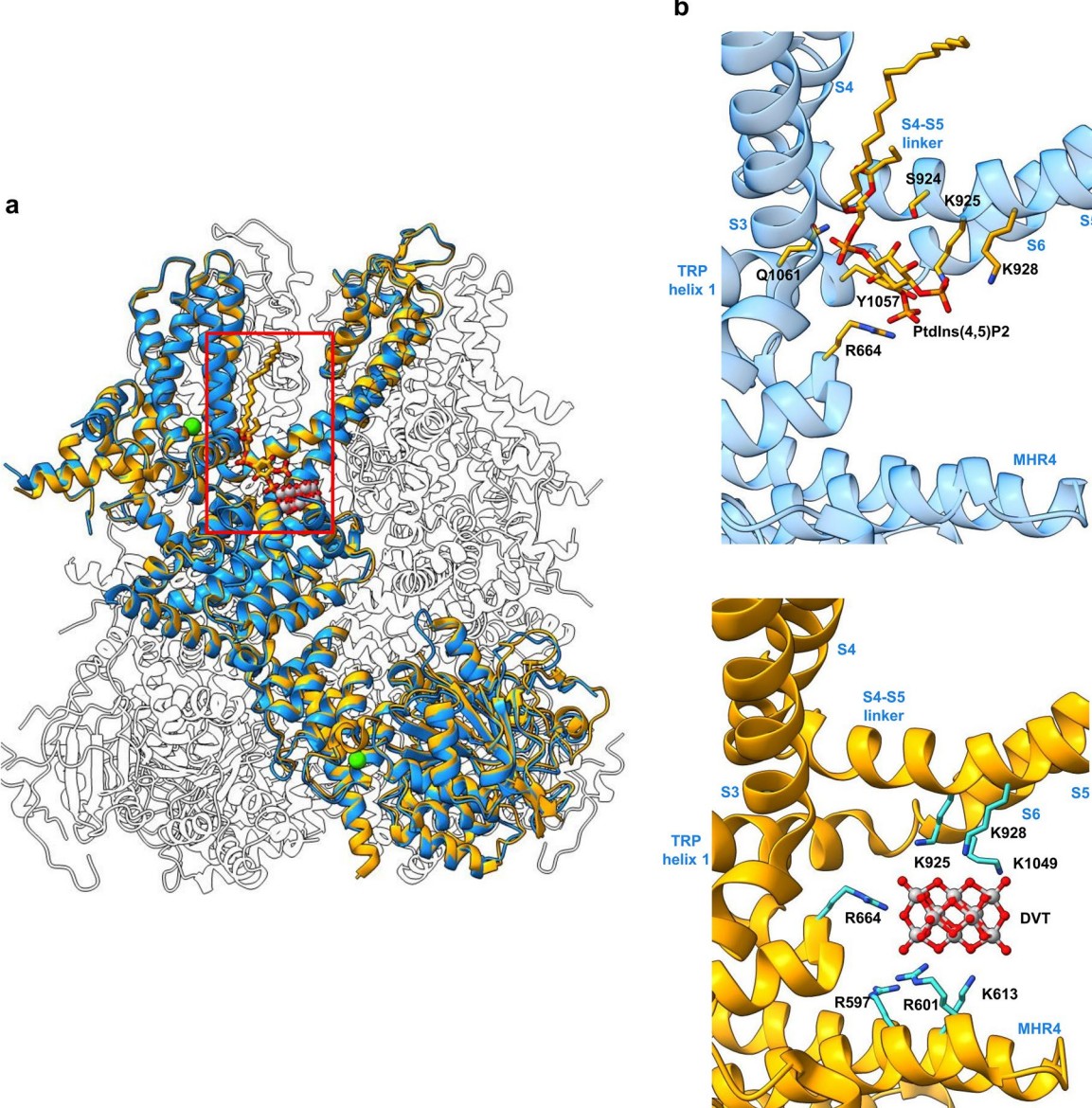

**Extended Data Fig. 6 | Structural comparison of the Ca²⁺/PtdIns(4,5)P₂-bound open TRPM4 (this study) and Ca²⁺/DVT-bound open TRPM4 at 37 °C (PDB: 9B8Y). a** Superposition of the two open state structures with the front subunit highlighted in blue and orange for the Ca²⁺/PtdIns(4,5)P₂-bound and Ca²⁺/ DVT$_{warm}$-bound structures, respectively. The region encompassing the PtdIns(4,5)P₂ and DVT binding sites is boxed. **b** Zoomed-in view of the PtdIns(4,5)P₂ (top) and DVT (bottom) binding sites with ligand-interacting residues shown in stick representation.

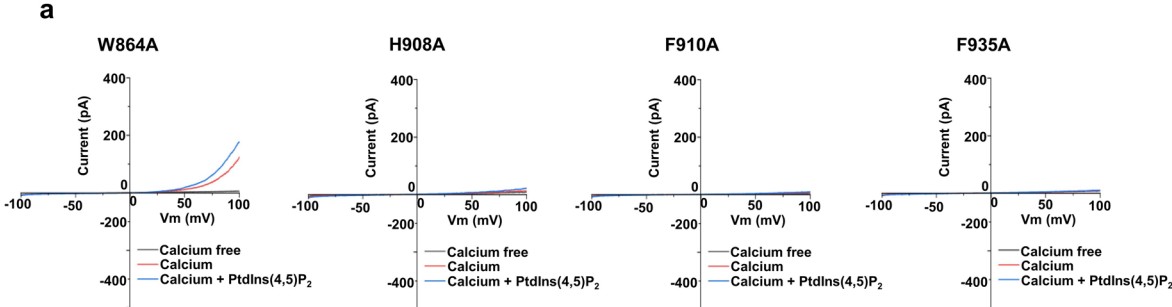

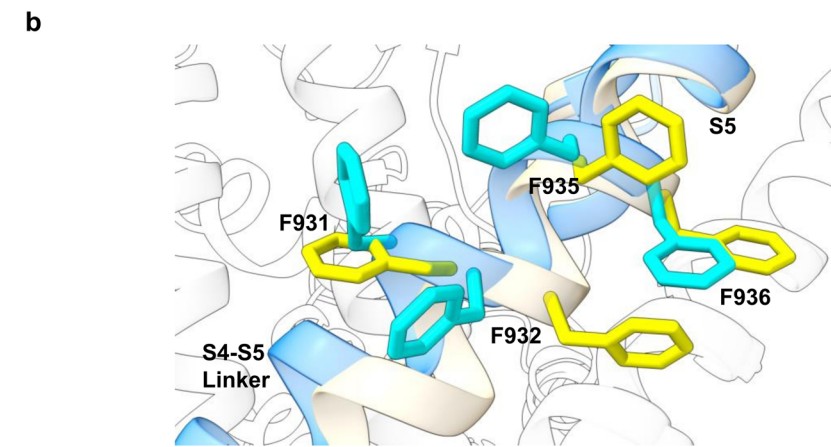

**Extended Data Fig. 7 | Ligand activation mechanism of hTRPM4. a** Sample I-V curves of hTRPM4 with mutations at the residues important for TRPM4 activation. Currents were recorded in inside-out patches with 300 µM Ca²⁺ or 300 µM Ca²⁺ and 10 µM PtdIns(4,5)P₂ di-C8 in the bath. Currents at +100 mV with the presence of Ca²⁺ and PtdIns(4,5)P₂ were used to plot Fig. 3e. **b** Concurrent rotation movements of the phenylalanine residues surrounding F935 on S5 upon TRPM4 activation. The phenylalanine side chains are colored cyan or yellow for the open or closed states, respectively. S4-S5 linker and S5 are highlighted in blue for the open state and wheat for the closed state.

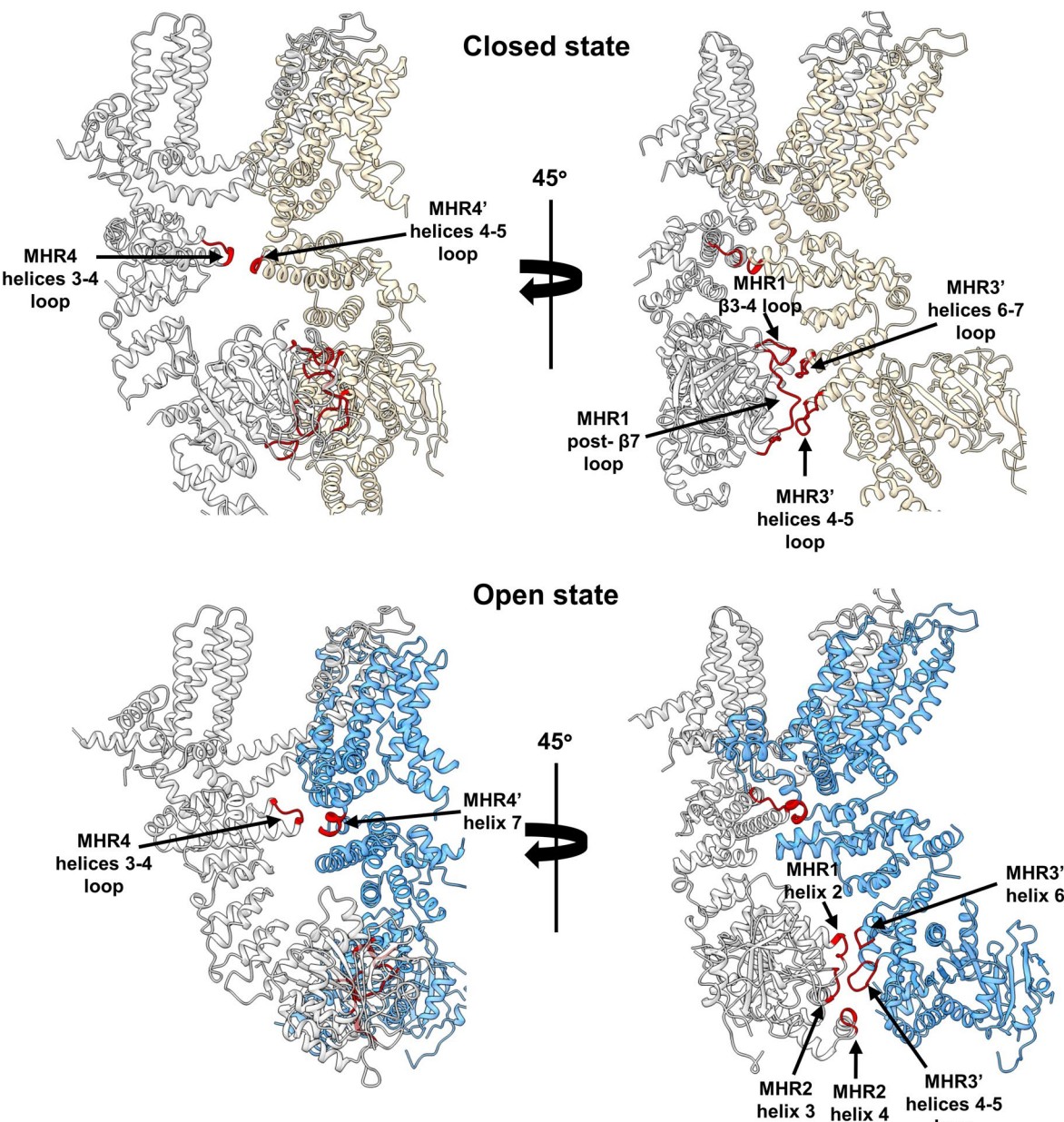

**Extended Data Fig. 8 | Changes of inter-subunit contacts at the MHR domains between closed and open states.** Two neighboring hTRPM4 subunits are shown in cartoon representation with the inter-subunit contact regions colored red. In the closed state (top), the MHR domains make two loose inter-subunit contacts: one contact, proximal to the membrane, is between the loop connecting helices 3 & 4 of MHR4 and the loop connecting helices 4 & 5 in the neighboring MHR4; the other contact, distal to the membrane, is between loops in MHR1 (β3-4 loop and post-β7 loop) and loops in MHR3 from the neighboring subunit (loops between helices 4 & 5 and between helices 6 & 7). In the open state (bottom), the upward swing of the MHR domains renders a new membrane-proximal contact between the turn of helices 3 & 4 from MHR4 and helix 7 of the neighboring MRH4 and a more extensive membrane-distal contact between the MHR1 and MHR2 domains (N-terminus of MHR1 helix 2, N-terminus of MHR2 helix 3 and C-terminus of MHR2 helix4) and the MHR3 domain from the neighboring subunit (C-terminus of helix 6 and the loop between helices 4 and 5).

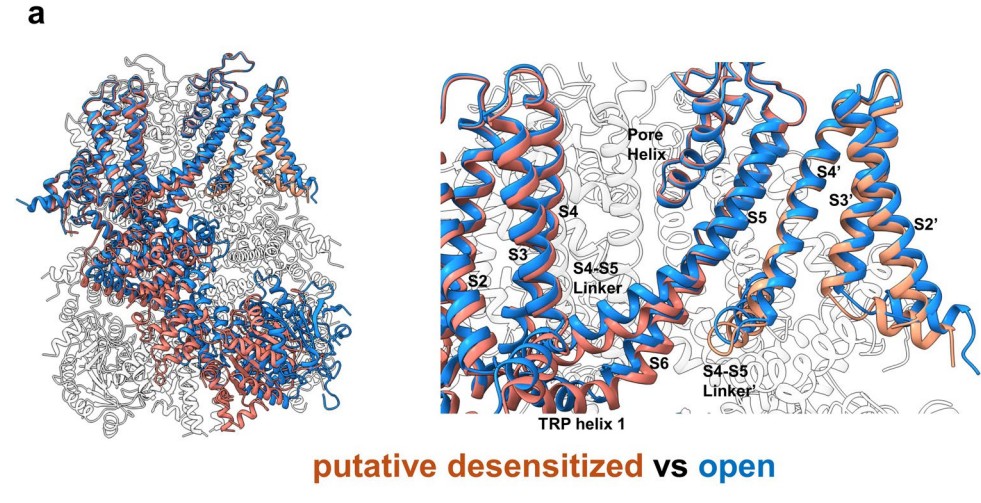

**putative desensitized vs open**

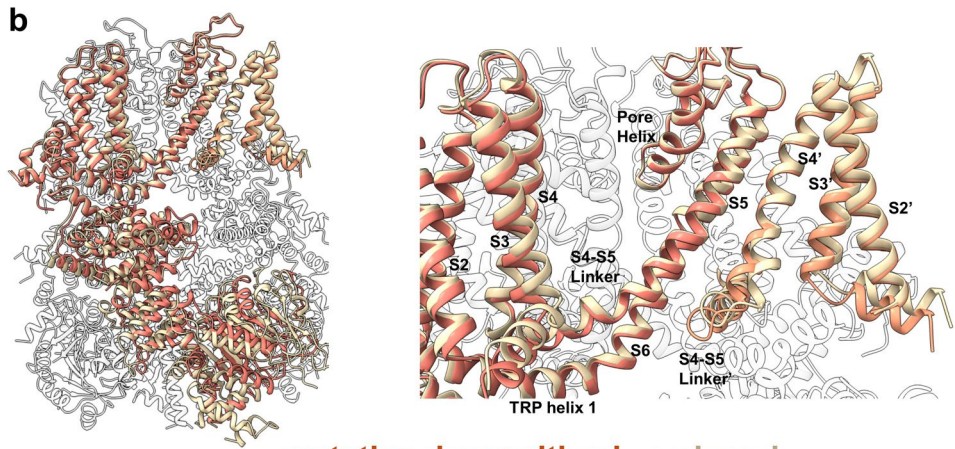

**putative desensitized vs closed**

**Extended Data Fig. 9 | Structural comparison of hTRPM4 in putative desensitized, open, and closed states. a** Structural comparison of TRPM4 between putative desensitized and open states. The front subunit and its neighboring S1-S4 domain are colored salmon in the desensitized state and blue in the open state. The right panel provides an enlarged view of the pore domain and its neighboring S1-S4. **b** Structural comparison of TRPM4 between putative desensitized and closed states. The front subunit and its neighboring S1-S4 domain are colored wheat in the closed state.

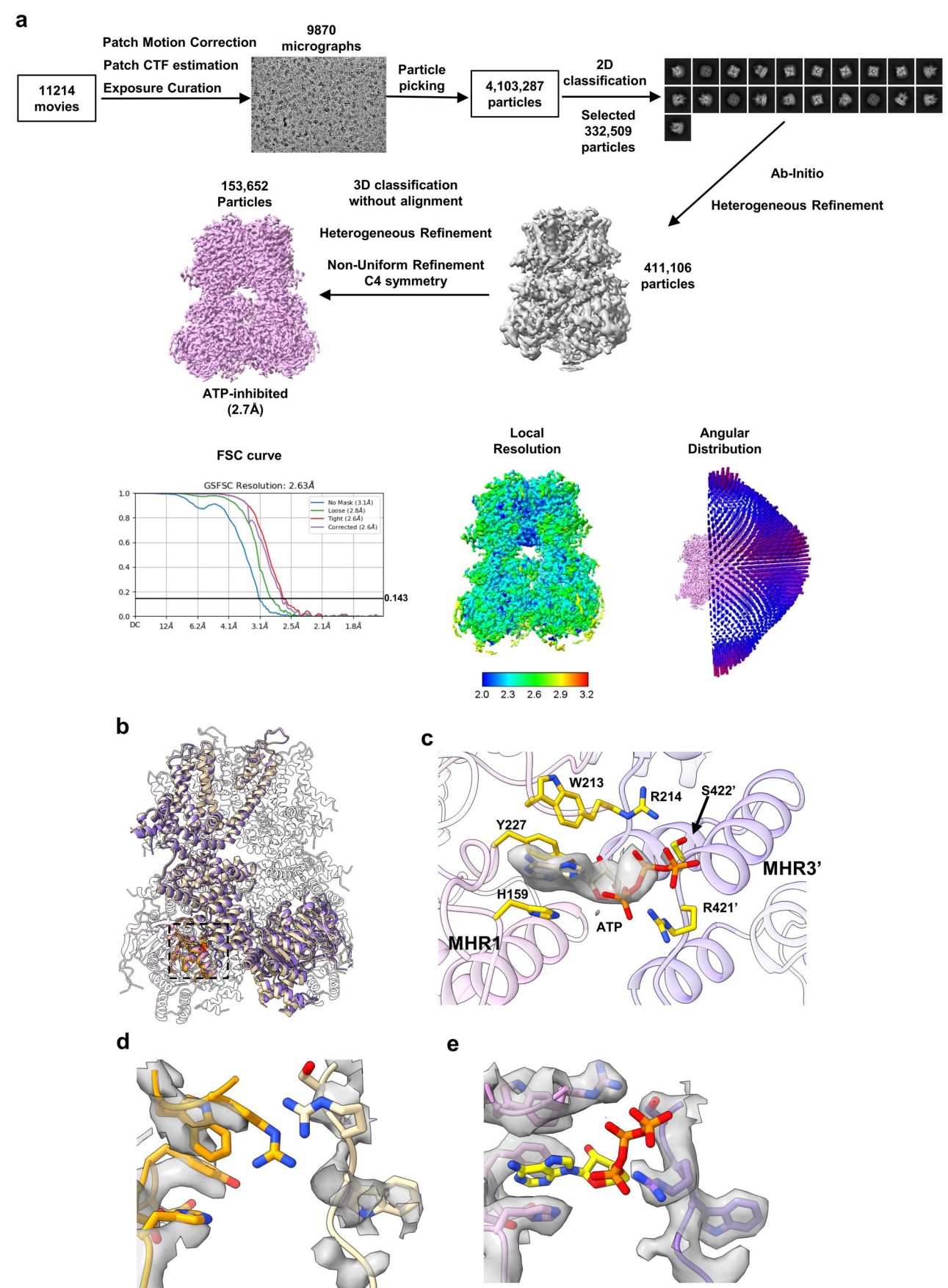

**Extended Data Fig. 10 | See next page for caption.**

**Extended Data Fig. 10 | Structure determination of ATP-inhibited hTRPM4.**
**a** Cryo-EM data processing scheme for the hTRPM4 sample prepared in the presence of Ca²⁺, PtdIns(4,5)P₂, and ATP. **b** Superposition of the hTRPM4 structures in apo closed and ATP-bound inhibited states. The front subunit is highlighted in purple for the ATP-bound state and wheat for the apo closed state. **c** Zoomed-in view of the ATP-binding site with the ATP density shown in grey surface contoured at 0.14 in ChimeraX. **d** Poorly defined side-chain density for those ATP-interacting residues (shown in sticks) in the EM map of the apo closed structure. **e** The side-chain density for the ATP-interacting residues is well defined in the EM map of the ATP-bound inhibited structure. Maps at the ATP-binding pocket shown in **d** and **e** are contoured at 0.12 in ChimeraX.

# Reporting Summary

## Statistics

For all statistical analyses, confirm that the following items are present in the figure legend, table legend, main text, or Methods section.

| n/a | Confirmed | |
|---|---|---|
| ☐ | ☒ | The exact sample size (*n*) for each experimental group/condition, given as a discrete number and unit of measurement |
| ☐ | ☒ | A statement on whether measurements were taken from distinct samples or whether the same sample was measured repeatedly |
| ☐ | ☒ | The statistical test(s) used AND whether they are one- or two-sided<br>*Only common tests should be described solely by name; describe more complex techniques in the Methods section.* |
| ☒ | ☐ | A description of all covariates tested |
| ☒ | ☐ | A description of any assumptions or corrections, such as tests of normality and adjustment for multiple comparisons |
| ☐ | ☒ | A full description of the statistical parameters including central tendency (e.g. means) or other basic estimates (e.g. regression coefficient) AND variation (e.g. standard deviation) or associated estimates of uncertainty (e.g. confidence intervals) |
| ☐ | ☒ | For null hypothesis testing, the test statistic (e.g. *F*, *t*, *r*) with confidence intervals, effect sizes, degrees of freedom and *P* value noted<br>*Give P values as exact values whenever suitable.* |
| ☒ | ☐ | For Bayesian analysis, information on the choice of priors and Markov chain Monte Carlo settings |
| ☒ | ☐ | For hierarchical and complex designs, identification of the appropriate level for tests and full reporting of outcomes |
| ☒ | ☐ | Estimates of effect sizes (e.g. Cohen's *d*, Pearson's *r*), indicating how they were calculated |

*Our web collection on statistics for biologists contains articles on many of the points above.*

## Software and code

Policy information about availability of computer code

| Data collection | SerialEM 3.8 |
|---|---|
| Data analysis | CryoSPARC 4.4.1 ; Coot 0.9.6; Phenix 1.21.2; UCSF ChimeraX 1.9; Graphpad Prism 8; HOLE 2.3.1; pClamp 11; Origin 2021b |

For manuscripts utilizing custom algorithms or software that are central to the research but not yet described in published literature, software must be made available to editors and reviewers. We strongly encourage code deposition in a community repository (e.g. GitHub). See the Nature Portfolio guidelines for submitting code & software for further information.

## Data

Policy information about availability of data

All manuscripts must include a data availability statement. This statement should provide the following information, where applicable:
- Accession codes, unique identifiers, or web links for publicly available datasets
- A description of any restrictions on data availability
- For clinical datasets or third party data, please ensure that the statement adheres to our policy

The cryo-EM density maps of the human TRPM4 have been deposited in the Electron Microscopy Data Bank (EMDB) under accession numbers EMD-48563 (Ca2+/PIP2-Open), EMD-48603 (Ca2+-Putative desensitized), EMD-48604 (Apo-Closed) and EMD-48605 (ATP-Inhibited). Atomic coordinates have been deposited in the Protein Data Bank (PDB) under accession numbers 9MRT (Ca2+/PIP2-Open), 9MT8 (Ca2+-Putative desensitized), 9MTA (Apo-Closed), and 9MTC (ATP-Inhibited).

# Research involving human participants, their data, or biological material

Policy information about studies with human participants or human data. See also policy information about sex, gender (identity/presentation), and sexual orientation and race, ethnicity and racism.

| | |
|---|---|
| Reporting on sex and gender | N/A |
| Reporting on race, ethnicity, or other socially relevant groupings | N/A |
| Population characteristics | N/A |
| Recruitment | N/A |
| Ethics oversight | N/A |

Note that full information on the approval of the study protocol must also be provided in the manuscript.

# Field-specific reporting

Please select the one below that is the best fit for your research. If you are not sure, read the appropriate sections before making your selection.

☒ Life sciences  ☐ Behavioural & social sciences  ☐ Ecological, evolutionary & environmental sciences

For a reference copy of the document with all sections, see nature.com/documents/nr-reporting-summary-flat.pdf

# Life sciences study design

All studies must disclose on these points even when the disclosure is negative.

| | |
|---|---|
| Sample size | The data size for cryo-EM was determined by the availability of the microscope time and the particle density on the grids. Sufficient cryo-EM datasets were collected to achieve the reported map resolution, which is sufficient for model building. |
| Data exclusions | CryoEM data processing involved removing poor-quality or damaged particles to achieve high resolution maps through pre-established standard data classification procedures. |
| Replication | For functional analysis,each experiment was reproduced at least two times on separate occasions. Experimental findings were reliably. reproduced. |
| Randomization | Randomization was not necessary as the independent variables to be tested were sufficient for the functional interpretations within this study. i.e. WT vs mutant current recordings |
| Blinding | All experiments are not blind. Blinding is not necessary for the purposes of structural determination. For functional analysis, blinding was not necessary due to the quantitative nature of the experiment. |

# Reporting for specific materials, systems and methods

We require information from authors about some types of materials, experimental systems and methods used in many studies. Here, indicate whether each material, system or method listed is relevant to your study. If you are not sure if a list item applies to your research, read the appropriate section before selecting a response.

## Materials & experimental systems

| n/a | Involved in the study |
|---|---|
| ☒ | Antibodies |
| ☐ | ☒ Eukaryotic cell lines |
| ☒ | Palaeontology and archaeology |
| ☒ | Animals and other organisms |
| ☒ | Clinical data |
| ☒ | Dual use research of concern |
| ☒ | Plants |

## Methods

| n/a | Involved in the study |
|---|---|
| ☒ | ChIP-seq |
| ☒ | Flow cytometry |
| ☒ | MRI-based neuroimaging |

## Eukaryotic cell lines

Policy information about cell lines and Sex and Gender in Research

| | |
|---|---|
| Cell line source(s) | Human embryonic kidney cells (HEK293 and HEK293S GnTI-) were purchased from ATCC and Sf9 insect cells were purchased from Thermo Fisher Scientific |
| Authentication | No further authentication was performed for commercially available cell lines |
| Mycoplasma contamination | Cell lines were not tested for mycoplasma contamination |
| Commonly misidentified lines (See ICLAC register) | None |

## Plants

| | |
|---|---|
| Seed stocks | N/A |
| Novel plant genotypes | N/A |
| Authentication | N/A |

