## [Peer Review File · Nature Structural & Molecular Biology]

Structural landscape of activation, desensitization, and inhibition in the human TRPM4 channel

Corresponding Author: Professor Youxing Jiang

A version of this paper was originally rejected for publication by Nature Structural & Molecular Biology, however that decision was reconsidered after appeal by the authors.

Version 0:

Decision Letter:

25th Feb 2025

Dear Dr. Jiang,

Thank you for submitting your manuscript "Structural landscape of activation, desensitization and inactivation in the human TRPM4 channel". I apologize for the delay in this decision. I'm writing to let you know that we have decided to send your manuscript for peer review.

I am re-opening the manuscript submission link for you to resubmit your manuscript with all the associated files needed for the peer review process directly to our system, at your convenience. Please see below for details regarding the required materials. Please follow the link at the bottom of this email to upload the documents.

We want to ensure that the methods and statistics reporting in our papers are of the highest quality. To that end, we ask authors to fill out a Reporting Summary that collects information on experimental design and reagents, as well as an editorial Policy Checklist, which confirms compliance with our editorial policies, including the declaration of Competing Interests. If your paper includes ChIP-seq, flow cytometry or MRI data, we ask you take special care to complete those sections of the Reporting Summary as this data will aid greatly in the review of your manuscript.

These documents can be found by following the links below:

Reporting Summary:

Editorial Policy Checklist: <https://www.nature.com/documents/nr-editorial-policy-checklist.pdf>

Please be aware of our guidelines on digital image standards.

IMPORTANT

In order for us to proceed with the peer review process of your manuscript, we require you to provide accession numbers and reviewer tokens to access sequencing data sets. Please add this information to your manuscript file.

Please note we require official wwPDB validation reports for newly described atomic structures, as noted in the policy checklist. We also request that authors provide cryo-EM maps, half-maps and models, to help the reviewers in assessing the work. We recommend the use of figshare integration into our systems, which allows for provision of anonymous access links for the referees (<https://www.springernature.com/gp/authors/research-data/figshare-integration>). Alternatively, please upload .zip folders directly with the submission. To ensure the ease of reviewer access to the data, please specify in the Data Availability section, where the files can be found (provide a figshare link or direct the reader to the manuscript files).

Additionally, I would like to kindly request that you provide the code used to analyse the data to the reviewers, if used. In order for the reviewers to evaluate the work adequately they must be able to test the software/review the code themselves. If you have not yet provided the software, we therefore request that you provide a single compressed zip file containing the

software with a readme.txt file or other user manual containing complete instructions for installing and running the software. If appropriate, please also provide example data and expected output. Sufficient material should be provided for referees to directly test the performance of the software/algorithm. If the software and materials are small enough to fit in a single compressed zip file less than 6MB in size, you may email this file directly to me. If the zip file is between 6 MB and 200 MB you may upload it to our file transfer site. If necessary, a second zip file up to 200 MB in size can be used to supply the example data. Please let me know if you need to use this option and I'll send you further details. Alternatively, you can also upload the code to GitHub and provide us with the link.

Please also fill out and return to me the code and software submission checklist that will be made available to editors and reviewers during manuscript assessment. Please note that this form is a dynamic 'smart pdf' and must therefore be downloaded and completed in Adobe Reader, instead of opening it in a web browser.

<https://www.nature.com/documents/nr-software-policy.pdf>

Please use the link below to submit the files. **Please also remember to move forward all other files associated with this version of the paper.**

Link Redacted

Sincerely,
Kat

Katarzyna Ciazynska, PhD
(she/her)
Senior Editor
Nature Structural & Molecular Biology
<https://orcid.org/0000-0002-9899-2428>

Version 1:

Decision Letter:

2nd Apr 2025

Dear Dr. Jiang,

Thank you again for submitting your manuscript "Structural landscape of activation, desensitization and inactivation in the human TRPM4 channel". [I apologize for the delay in responding, which resulted from the difficulty in obtaining suitable referee reports. Nevertheless,] we now have comments (below) from the 2 reviewers who evaluated your paper. In light of those reports, we remain interested in your study and would like to see your response to the comments of the referees, in the form of a revised manuscript.

You will see that while reviewers appreciate the results, they raise several concerns which will need to be addressed in a revision. Specifically, both reviewers request a more in depth comparison with previously published data, including other TRP channels. Additionally, we agree with reviewer #1 that revisions on statistical reporting will make the manuscript more robust, together with confirmation of the functionality of the natural PIP2 used compared to DiC8 PIP2. We ask, in line with reviewer #2 comments, that you attempt to identify the putative calcium density and provide more discussion on its role. We also ask that you experimentally address point 3 of reviewer #2, pertaining to desensitized state reported, and that you consider the reviewer points about the ATP bound state, performing additional experiments when and as needed.

Please be sure to address/respond to all concerns of the referees in full in a point-by-point response and highlight all changes in the revised manuscript text file. If you have comments that are intended for editors only, please include those in a separate cover letter.

We expect to see your revised manuscript within 6 weeks. If you cannot send it within this time, please contact us to discuss an extension; we would still consider your revision, provided that no similar work has been accepted for publication at NSMB or published elsewhere.

Reporting Summary:

EXTENDED DATA FIGURES

Please note that all key data shown in the main figures as cropped gels or blots should be presented in uncropped form, with molecular weight markers. These data can be aggregated into a single supplementary figure item. While these data can be displayed in a relatively informal style, they must refer back to the relevant figures. These data should be submitted with the final revision, as source data, prior to acceptance, but you may want to start putting it together at this point.

Data availability: this journal strongly supports public availability of data. All data used in accepted papers should be available via a public data repository, or alternatively, as Supplementary Information. If data can only be shared on request, please explain why in your Data Availability Statement, and also in the correspondence with your editor. Please note that for some data types, deposition in a public repository is mandatory - more information on our data deposition policies and available repositories can be found below:

<https://www.nature.com/nature-research/editorial-policies/reporting-standards#availability-of-data>

Nature Structural & Molecular Biology is committed to improving transparency in authorship. As part of our efforts in this direction, we are now requesting that all authors identified as 'corresponding author' on published papers create and link their Open Researcher and Contributor Identifier (ORCID) with their account on the Manuscript Tracking System (MTS), prior

to acceptance. This applies to primary research papers only. ORCID helps the scientific community achieve unambiguous attribution of all scholarly contributions. You can create and link your ORCID from the home page of the MTS by clicking on 'Modify my Springer Nature account'. For more information please visit www.springernature.com/orcid.

Link Redacted

Sincerely,

Katarzyna Ciazynska, PhD
(she/her)
Senior Editor
Nature Structural & Molecular Biology
<https://orcid.org/0000-0002-9899-2428>

Referee expertise:

Referee #1: TRP channels, molecular biology

Referee #2: TRP channels, cryoEM

Reviewers' Comments:

Reviewer #1 (Remarks to the Author):

The manuscript by Teixeira-Duarte and colleagues studies the Ca²⁺ activated non-selective cation channel TRPM4, using structural approaches as well as patch clamp electrophysiology. They determine the structure of TRPM4 in the presence of its co-factor PIP2 and its activator Ca²⁺. The cryoEM data shows three populations, apo-closed, Ca²⁺ and PIP2 bound-open and bound to only Ca²⁺ - presumably desensitized. They also determine the structure of TRPM4 in the presence of ATP, which they designate as inactivated state. In this reviewer's opinion these novel structures provide very important contribution, and they warrant publication in NSMB. Specifically, PIP2 is a general co-factor for most TRP channels, which includes all members of the TRPM sub-family, and there are only a limited number of structures where this lipid is bound to the channel, and even less where the channel is in an open conformation. Also, essentially all earlier TRP channel structures with PIP2 used the synthetic short acyl chain variant (DiC8) (see PMID37871124 for list/review), whereas the current manuscript used the natural long acyl chain variant – this by itself makes it an important contribution. I have only a few relatively minor comments to improve the presentation of the manuscript and to put into the context of the regulation of other TRP channels by PIP2.

1. The papers describing structures of TRPV5, TRPM8 and TRPM3 together with PIP2 should be cited and discussed. Briefly: TRPV5, a constitutively active Ca²⁺ channel, shows an open structure in the presence of diC8 PIP2 (PMID: 35476976, 30305626). TRPM8, the cold- and menthol-activated channel is not open in the combined presence of diC8 PIP2 and its chemical agonist (PMID: 30733385), and only a combination of two different kinds of agonists plus PI(4,5)P2 opens it (PMID: 36227998). TRPM3, also did not show an open structure in the presence of diC8 PI(4,5)P2 and its chemical agonist PregS (PMID: 36283409). The authors may also consider discussing the literature, including their own work, on structures of TRPML1 with PI(3,5)P2.
2. The authors mention that the binding site for PI(4,5)P2 in TRPM4, is different from those in TRPM3 and TRPM8, but similar to those in TRPV1 and TRPV5. Showing the PI(4,5)P2 interacting residues in these channel in a multiple sequence alignment would be very useful for the readers. Also, I think PI(4,5)P2 binds to TRPV5 and TRPV1 at different places.
3. Figure 1A. The excised patch data is nice to illustrate the effect of PI(4,5)P2, and even though it is not novel, showing some statistical summary would be desirable. Also, this measurement is performed with DiC8 PI(4,5)P2, but the structure is determined with long acyl chain natural PI(4,5)P2 purified from brain. It would be important to demonstrate the functional effect of natural PI(4,5)P2 in excised patches, given that all previously published functional data on TRPM4 were obtained using DiC8 PI(4,5)P2.

4. Figure 3: the pore profiles show the difference between the closed and open states. It would be informative to also show the pore profiles of the desensitized (Ca²⁺ only) and inactivated (ATP bound) states, to show how much they differ from the closed state.

Additional minor comment:

5. Line 37 TRPM4 and TRPM5 share high sequence identity – I would specify how high the sequence identity is

Reviewer #2 (Remarks to the Author):

Teixeira-Duarte et al. presented cryo-EM structures of TRPM4 in its Ca²⁺ and PIP₂-bound open state, Ca²⁺-bound intermediate state, ATP-bound inhibited state, and apo state, revealing the conformational landscape of TRPM4 activation and inhibition.

The recent cryo-EM studies of TRPM4 at two different temperatures (22 °C and 37 °C) led to the proposal of a temperature-dependent change in the binding sites for Ca²⁺ and decavanadate (DVT), as well as the physiological temperature (37 °C) requirement for channel opening (Hu et al., 2024). Although intriguing, its open state was achieved with the assistance of an exogenous molecule (DVT), which is a highly negatively charged, symmetrical, spiky ball-like molecule. Therefore, the conformational landscape of TRPM4 activation in a physiologically relevant environment and its temperature-dependent ligand recognition and opening remain unclear.

This study demonstrated that PIP₂ is required to prevent desensitization and presented the Ca²⁺ and PIP₂-bound open state at 12 °C, rather than 37 °C. Furthermore, thorough cryo-EM studies provide a plausible structural analysis highlighting the mechanisms of Ca²⁺-dependent TRPM4 opening and ATP-dependent inhibition, further clarifying the mechanisms surrounding this system. Overall, the structural studies (cryo-EM and modeling) are well-executed, and the impact of this study is high.

Although my enthusiasm is high, there are major points that the authors must address before I can recommend publication to NSMB.

Major points.

1) It is critical to provide a detailed comparison between the current and previous studies to draw conclusions about the roles of PIP₂, Ca²⁺, temperature, and ATP in TRPM4 due to several contrasting features in the two studies. However, the authors only briefly commented on previous findings and did not provide such a comparison. I quickly compared the open state with Ca²⁺/PIP₂/12C to that of Ca²⁺/DVT/37C and found that they are quite similar in their conformations. The previous studies highlighted the significance of physiological temperature for TRPM4 opening, but this study contrasts with that, likely because the physiological modulator PIP₂ is employed. I recommend making comparisons in a main figure for the following sites: 1) PIP₂ versus DVT sites, 2) the intracellular Ca²⁺ site, and 3) overall conformations and discussions.

2) I realized the previous study observed an intracellular Ca²⁺ site in the open state. Inspection of the cryo-EM map of the Ca²⁺/PIP₂ open state has a density, possibly corresponding to Ca²⁺. However, the authors did not model that as the Ca²⁺ site. Is there a reason not to model this as Ca²⁺? If so, please provide the reasoning and data in the manuscript.

3) The authors claim the Ca²⁺-bound conformation is the Ca²⁺ desensitized state, based solely on electrophysiological characterization. However, as the authors acknowledge, the Ca²⁺-bound conformation is an intermediate between the apo and Ca²⁺/PIP₂-open states, suggesting it may represent an intermediate state. Furthermore, I observed that the intracellular putative Ca²⁺ is absent in this Ca²⁺-bound state. I would expect the desensitized state to favor Ca²⁺ binding at both sites because the channel must open before desensitization occurs. Please provide functional data and/or structural analysis to support this conformation as a desensitized state and to rule out the possibility that this is an intermediate state.

4) The authors claim that the ATP-bound state is an ATP-inactivated state, perhaps because the authors included both Ca²⁺, PIP₂, and ATP. However, it is possible that in most cases, ATP binds to the apo state first and then stabilizes the apo state in the sample. Consistent with this scenario, the MHR arrangement of the ATP-bound state looks very similar to the apo state, and the role of ATP seems to strengthen the inter-subunit interactions of MHRs (MHR1 and MHR3), as stated by the authors. In my opinion, the ATP-bound state represents the ATP-inhibited, and its role is to stabilize the closed state. For this reason, I suggest changing the definition of the state as “ATP-inhibited state” unless the authors have functional data to support that this conformation as an inactivated (deactivated? Why inactivated?) state.

5) Finally, I noticed that the current ATP-inactivated state lacks Ca²⁺ in S1-S4, which is interesting, while the previous study appears to have Ca²⁺ in S1-S4. Additionally, the previous study indicates a different ATP binding site at 37 °C compared to 22 °C. Please compare the current ATP-bound structure with the previously published structures in a revision.

Minor points

1) Line 148: please provide references for TRPV1, TRPV5, TRPM3, and TRPM8 studies.

2) Line 155-156: Based on the structural analysis, the authors conclude that Ca²⁺ binds and activates first then PIP₂ binds.

Although it is logical based on the conformational selection mechanism, it is also possible that initial PIP2 binding can induce conformation for Ca²⁺ binding or synergistic Ca²⁺ and PIP2 binding (induced fit type). Also, it seems the authors purify protein with PIP2 and then add Ca²⁺, so PIP2 should bind to protein at least loosely first. Thus, I suggest the authors tone down this sentence, acknowledging the other possibilities unless the authors have functional data.

Version 2:

Decision Letter:

Nature Structural & Molecular Biology NSMB-A50530B

16th May 2025

Dear Dr. Jiang,

Thank you for submitting your manuscript, "Structural landscape of activation, desensitization and inhibition in the human TRPM4 channel". The comments of 2 expert referees are below. You will see that the reviewers have persistent serious concerns, which will preclude us from further consideration of the manuscript. Specifically, reviewer #1 states that the efforts made in revision to address their comments about PIP2 modulations are not adequate. Reviewer #2, while appreciative of the work, states that the conclusions on the conformational states are not fully convincing. In light of these persistent concerns and remaining open questions, we cannot offer to publish the study in Nature Structural & Molecular Biology. We hope the referees' comments will be useful to you in revising the manuscript for submission elsewhere.

However, considering the strengths of the manuscript, we took the liberty to consult our colleagues at Nature Communications and EMBO Journal. Please see below for the offers for further consideration. To transfer the manuscript, please follow the link in the footnote, which will allow you to choose a journal to transfer to.

The editors at Nature Communications are interested in the work and, depending on the extent of your revisions, will either send the revised manuscript back to reviewers or offer acceptance in principle. If you have any queries, or would like to discuss the extent of revisions necessary, please contact Katarzyna Marcinkiewicz (katarzyna.marcinkiewicz@us.nature.com) who will be the handling editor there.

While we cannot continue with this manuscript at our journal, we have discussed the study with our colleagues at The EMBO Journal. As you may know, The EMBO Journal is a non-profit, fully open-access journal published by EMBO Press and offers rapid and transparent peer review with a single major revision round policy. The handling editor Ieva Gailite would be happy to consider your manuscript for publication at their journal after extending the discussion along the lines indicated by both reviewers. Furthermore, Ieva would like to discuss with you the feasibility of including dipalmitoyl PIP2 data either via email or a Zoom call.

If you are interested in this option, please submit the revised manuscript to The EMBO Journal via the transfer link below, while indicating its history in the cover letter. Reformatting is not needed at this point. Please feel free to contact Ieva at i.gailite@embojournal.org if you have any questions about the transfer, The EMBO Journal or its policies.

I am sincerely sorry we could not be more positive on this occasion.

Sincerely,

Katarzyna Ciazynska, PhD
(she/her)
Senior Editor
Nature Structural & Molecular Biology
<https://orcid.org/0000-0002-9899-2428>

Reviewer #1 (Remarks to the Author):

While the manuscript is an important contribution, I was somewhat disappointed that the authors did not put appropriate effort into performing a simple, but important requested experiment demonstrating the effect of native long acyl chain PIP2 on TRPM4. While natural PIP2 is somewhat more difficult to handle than DiC8 PIP2, there are ample reports in the literature on PIP2 and ion channels from many labs, including early papers from the Hilgemann, Nichols and Logothetis labs showing clear effects of natural PIP2 on ion channels in excised inside out patches. After sonication in an aqueous buffer, this lipid was shown to activate a wide variety of PIP2 sensitive ion channels. The effect is usually slower than that of DiC8 PIP2, but on

most channels it requires lower concentrations and gives a larger effect than DiC8 PIP2. I think this is a critical experiment, and the authors should put appropriate effort into performing it and including in in the manuscript. Dipalmitoyl PIP2 is an alternative, if natural PIP2 remains too problematic, it is more soluble than native PIP2, but it is a lot closer than DiC8 PIP2 to the natural form.

I also think the authors should discuss at least some of the structures on other TRP(M) channels and PIP2. I do not believe it would be out of place. The current discussion of the manuscript is very short, and such discussion, even if brief, would make the manuscript more accessible to a wider readership.

Reviewer #2 (Remarks to the Author):

The revised manuscript has addressed most of my concerns and is improved. I appreciate the authors' structural analysis of the Ca²⁺-bound closed state and their rationale for designating it as a desensitized state. However, as the authors acknowledge, it remains equally plausible that this conformation represents an intermediate state rather than a desensitized one. Given that Ca²⁺ binding/activation initiates in the S1–S4 then propagates to the pore regions, the closed conformation of the MHR domain in the absence of bound Ca²⁺ is also consistent with an intermediate state. Alternatively, if this conformation reflects a desensitized state, the authors will need to invoke Ca²⁺ release from the MHRs after channel opening—an energetically challenging scenario that is not convincingly supported. Regardless, this is an important study, and I do not want to delay its publication. To proceed without further revision, I suggest that the authors discuss both possibilities—the intermediate and desensitized interpretations—and provide a rationale for favoring the desensitized state. Importantly, the authors should refer to this conformation as a putative desensitized state throughout the manuscript, which would be more accurate based on the current data. Again, congratulations on this milestone study on the mechanism of TRPV4 activation.

** As a service to authors, Springer Nature Limited provides authors with the ability to transfer a manuscript that one journal cannot offer to publish to another journal, without the author having to upload the manuscript data again. To transfer your manuscript to another NPG journal using this service, please click on Link Redacted

** For Springer Nature Limited general information and news for authors, see <http://npg.nature.com/authors>.

Version 3:

Decision Letter:

Our ref: NSMB-A50530C-Z

21st Aug 2025

Dear Dr. Jiang,

Thank you for submitting your revised manuscript "Structural landscape of activation, desensitization, and inhibition in the human TRPM4 channel" (NSMB-A50530C-Z). It has now been seen by the original referees and their comments are below. The reviewers find that the paper has improved in revision, and therefore we'll be happy in principle to publish it in Nature Structural & Molecular Biology, pending minor revisions to satisfy the referees' final requests and to comply with our editorial and formatting guidelines.

We are now performing detailed checks on your paper and will send you a checklist detailing our editorial and formatting requirements in about 2-3 weeks. Please do not upload the final materials and make any revisions until you receive this additional information from us.

To facilitate our work at this stage, it is important that we have a copy of the main text as a word file. If you could please send along a word version of this file as soon as possible, we would greatly appreciate it; please make sure to copy the NSMB account (cc'ed above).

Sincerely,

Katarzyna Ciazynska, PhD
(she/her)
Senior Editor

Nature Structural & Molecular Biology
<https://orcid.org/0000-0002-9899-2428>

Reviewer #1 (Remarks to the Author):

I recommend acceptance

Reviewer #2 (Remarks to the Author):

The authors addressed previous points in this revision. Again, this is an important study and I fully recommend its publication in NSMB.

Version 4:

Decision Letter:

1st Oct 2025

Dear Dr. Jiang,

We are now happy to accept your revised paper "Structural landscape of activation, desensitization, and inhibition in the human TRPM4 channel" for publication as an Article in Nature Structural & Molecular Biology.

Your paper will be published online soon after we receive proof corrections and will appear in print in the next available issue. You can find out your date of online publication by contacting the production team shortly after sending your proof corrections.

Authors may need to take specific actions to achieve compliance with funder and institutional open access mandates. If your research is supported by a funder that requires immediate open access (e.g. according to [Plan S principles](https://www.springernature.com/gp/open-science/plan-s-compliance) or the [NIH public access policy](https://www.springernature.com/gp/open-science/us-federal-agency-compliance)) then you should select the gold OA route, and we will direct you to the compliant route where possible. Because authors warrant under our subscription licensing terms that they haven't committed to licensing any version of their article under a licence inconsistent with the terms of our agreement – including the applicable embargo period – publication under the subscription model isn't suitable for authors whose funders require no embargo.

Sincerely,

Katarzyna Ciazynska, PhD
(she/her)
Senior Editor
Nature Structural & Molecular Biology
<https://orcid.org/0000-0002-9899-2428>

To Reviewer #1

The manuscript by Teixeira-Duarte and colleagues studies the Ca²⁺ activated non-selective cation channel TRPM4, using structural approaches as well as patch clamp electrophysiology. They determine the structure of TRPM4 in the presence of its co-factor PIP2 and its activator Ca²⁺. The cryoEM data shows three populations, apo-closed, Ca²⁺ and PIP2 bound-open and bound to only Ca²⁺ - presumably desensitized. They also determine the structure of TRPM4 in the presence of ATP, which they designate as inactivated state. In this reviewer's opinion these novel structures provide very important contribution, and they warrant publication in NSMB. Specifically, PIP2 is a general co-factor for most TRP channels, which includes all members of the TRPM sub-family, and there are only a limited number of structures where this lipid is bound to the channel, and even less where the channel is in an open conformation. Also, essentially all earlier TRP channel structures with PIP2 used the synthetic short acyl chain variant (DiC8) (see PMID37871124 for list/review), whereas the current manuscript used the natural long acyl chain variant – this by itself makes it an important contribution. I have only a few relatively minor comments to improve the presentation of the manuscript and to put into the context of the regulation of other TRP channels by PIP2.

We appreciate reviewer #1's positive comments and constructive suggestions. The points raised by the reviewer are addressed as follows:

1. The papers describing structures of TRPV5, TRPM8 and TRPM3 together with PIP2 should be cited and discussed. Briefly: TRPV5, a constitutively active Ca²⁺ channel, shows an open structure in the presence for diC8 PIP2 (PMID: 35476976, 30305626). TRPM8, the cold- and menthol-activated channel is not open in the combined presence of diC8 PIP2 and its chemical agonist (PMID: 30733385), and only a combination of two different kinds of agonists plus PI(4,5)P2 opens it (PMID: 36227998). TRPM3, also did not show an open structure in the presence of diC8 PI(4,5)P2 and its chemical agonist PregS (PMID: 36283409). The authors may also consider discussing the literature, including their own work, on structures of TRPML1 with PI(3,5)P2.

All the papers related to the structures of TRPV5, TRPV1, TRPM3, TRPM8 in complex with PIP2 have been referenced in the revision as suggested. We have also referenced the review paper (PMID37871124) mentioned by the reviewer which provides a nice summary of PIP2 regulation in various TRP channels. The suggested discussion about the structural information of all these TRP channels with PIP2 is a bit out of place in the main text. In the revision, we provided this information in the legend of Supplementary Figure 4 where we compare the PIP2-bound structures of those TRP channels.

2. The authors mention that the binding site for PI(4,5)P2 in TRPM4, is different from those in TRPM3 and TRPM8, but similar to those in TRPV1 and TRPV5. Showing the PI(4,5)P2 interacting residues in these channel in a multiple sequence alignment would be very useful for the readers. Also, I think PI(4,5)P2 binds to TRPV5 and TRPV1 at different places.

The comparison of PIP2 sites in various TRP channels aims to demonstrate that PIP2 binding in TRPM4 is different from that in TRPM3 and 8. Because of the different PIP2 locations between TRPM4 and TRPM3/8, the key PIP2-interacting residues are positioned at different regions of the channels, making it difficult to make proper sequence alignment. Furthermore, although PIP2 binding in TRPM4 is similar to the PIP2 site at the vanilloid-binding pocket region of TRPV1 and TRPV5, the positions of PIP2-interacting residues are quite diverse. This, along with low sequence similarity between TRPM4 and TRPV channels, also makes it difficult to properly align their sequences. Thus, it would be challenging to align the sequences of PIP2 sites and compare the PIP2-interacting residues among these TRP channels.

3. Figure 1A. The excised patch data is nice to illustrate the effect of PI(4,5)P₂, and even though it is not novel, showing some statistical summary would be desirable. Also, this measurement is performed with DiC8 PI(4,5)P₂, but the structure is determined with long acyl chain natural PI(4,5)P₂ purified from brain. It would be important to demonstrate the functional effect of natural PI(4,5)P₂ in excised patches, given that all previously published functional data on TRPM4 were obtained using DiC8 PI(4,5)P₂.

As suggested, we have provided statistical analysis of the current levels at various states in response to Ca²⁺ and/or PIP2 (A to E states in Figure A). Due to the variation of the current sizes from patch to patch, the currents at different states were normalized against the maximum currents at the initial Ca²⁺-activated state (state A) in the analysis (Figure 1c in revision)

The functional effect of native long-chain PIP2 is difficult to measure due to its insolubility in water. We performed a pilot recording experiment to test the native brain PIP2 effect on TRPM4 activity (see figure below). Because of its insolubility, the long-chain lipid had to be prepared as a

liposome stock in water and applied to the perfusion solution in our recordings. We can still observe some potentiation effect from the native lipid. However, likely caused by the low effectiveness of the liposome fusion into the patch membrane, it takes a long time before seeing the potentiation effect of

the native lipid, and the effect is less potent than that from water-soluble diC8 PIP2. We did not include this data in revision, but explained the use of water-soluble diC8 PIP2 instead of long-chain native lipid in the Methods.

4. Figure 3: the pore profiles show the difference between the closed and open states. It would be informative to also show the pore profiles of the desensitized (Ca²⁺ only) and inactivated (ATP bound) states, to show how much they differ from the closed state.

As suggested, we have included the pore profiles of the desensitized and ATP-inhibited states in Figure 3f and 3g.

5. Line 37 TRPM4 and TRPM5 share high sequence identity – I would specify how high the sequence identity is.

As suggested, we have included the sequence identity value (~50%) in the revision.

To Reviewer #2

Teixeira-Duarte et al. presented cryo-EM structures of TRPM4 in its Ca²⁺ and PIP₂-bound open state, Ca²⁺-bound intermediate state, ATP-bound inhibited state, and apo state, revealing the conformational landscape of TRPM4 activation and inhibition.

The recent cryo-EM studies of TRPM4 at two different temperatures (22 °C and 37 °C) led to the proposal of a temperature-dependent change in the binding sites for Ca²⁺ and decavanadate (DVT), as well as the physiological temperature (37 °C) requirement for channel opening (Hu et al., 2024). Although intriguing, its open state was achieved with the assistance of an exogenous molecule (DVT), which is a highly negatively charged, symmetrical, spiky ball-like molecule. Therefore, the conformational landscape of TRPM4 activation in a physiologically relevant environment and its temperature-dependent ligand recognition and opening remain unclear.

This study demonstrated that PIP₂ is required to prevent desensitization and presented the Ca²⁺ and PIP₂-bound open state at 12 °C, rather than 37 °C. Furthermore, thorough cryo-EM studies provide a plausible structural analysis highlighting the mechanisms of Ca²⁺-dependent TRPM4 opening and ATP-dependent inhibition, further clarifying the mechanisms surrounding this system. Overall, the structural studies (cryo-EM and modeling) are well-executed, and the impact of this study is high.

Although my enthusiasm is high, there are major points that the authors must address before I can recommend publication to NSMB.

We would like to thank Review #2's positive critiques of our manuscript. The reviewer has provided some constructive suggestions, and the points raised by the reviewers are addressed as follows:

Major points.

1) It is critical to provide a detailed comparison between the current and previous studies to draw conclusions about the roles of PIP₂, Ca²⁺, temperature, and ATP in TRPM4 due to several contrasting features in the two studies. However, the authors only briefly commented on previous

findings and did not provide such a comparison. I quickly compared the open state with Ca²⁺/PIP2/12C to that of Ca²⁺/DVT/37C and found that they are quite similar in their conformations. The previous studies highlighted the significance of physiological temperature for TRPM4 opening, but this study contrasts with that, likely because the physiological modulator PIP2 is employed. I recommend making comparisons in a main figure for the following sites: 1) PIP2 versus DVT sites, 2) the intracellular Ca²⁺ site, and 3) overall conformations and discussions.

As suggested, we have included the following in the revision:

1. We have added the structural comparison between PIP2 and DVT sites (Supplementary Figure 5b) with a brief discussion in the main text after describing PIP2 binding.
2. We have added the figure and description of the intracellular Ca²⁺ site in the section related to Ca²⁺ binding (Figure 2d & 2e). It is worth noting that this intracellular site forms in the open state but not in the closed state due to conformational changes at the intracellular domains. Two of the surrounding residues (D270 and C385) at this Ca²⁺ site are far from the other two (D395 and E396) in the closed state. This would imply that Ca²⁺-binding at this site may help stabilize the open state but is not essential for initial channel activation. Intriguingly, among the three Ca²⁺-chelating residues (E395, E396, and D270), only the E396 mutation shows any effect on channel activity. This discussion is also included in the revision.
3. We have added the structural comparison between our open structure and the previously determined open structure at a high temperature (Supplementary Figure 5a). Both structures are very similar. As pointed out by this reviewer, our open structure was obtained at 12 °C and electrophysiological recordings were performed at room temperature, suggesting that the physiological temperature (37 °C) is not the determining factor for TRPM4 activation. We suspect that a higher temperature may enhance the dynamic movement of the intracellular domains and facilitate DVT access to its active site similar to where the PIP2 head group resides, which in turn helps stabilize the channel in the open state just like PIP2. We have included this brief discussion in the revision.

2) I realized the previous study observed an intracellular Ca²⁺ site in the open state. Inspection of the cryo-EM map of the Ca²⁺/PIP2 open state has a density, possibly corresponding to Ca²⁺. However, the authors did not model that as the Ca²⁺ site. Is there a reason not to model this as Ca²⁺? If so, please provide the reasoning and data in the manuscript.

As suggested, we have added the figure and description of the intracellular Ca²⁺ site in the revision (Figure 2d & 2e).

3) The authors claim the Ca²⁺-bound conformation is the Ca²⁺ desensitized state, based solely on electrophysiological characterization. However, as the authors acknowledge, the Ca²⁺-bound conformation is an intermediate between the apo and Ca²⁺/PIP2-open states, suggesting it may

represent an intermediate state. Furthermore, I observed that the intracellular putative Ca²⁺ is absent in this Ca²⁺-bound state. I would expect the desensitized state to favor Ca²⁺ binding at both sites because the channel must open before desensitization occurs. Please provide functional data and/or structural analysis to support this conformation as a desensitized state and to rule out the possibility that this is an intermediate state.

In the Ca²⁺-bound TRPM4 structure, the S1-S4 domain maintains the Ca²⁺-bound activated conformation, but its pore domain and entire intracellular domains are in the closed conformation due to the loss of PIP₂ stabilization, mimicking the condition of TRPM4 desensitization caused by the membrane PIP₂ depletion after Ca²⁺ activation. As discussed in our previous response, the intracellular Ca²⁺ site exists in the open state but not in the closed state due to conformational changes at the intracellular domains, which explains the lack of intracellular Ca²⁺ binding in this Ca²⁺-bound desensitized structure (in which the entire intracellular domains adopt closed conformation).

Our structural analysis of TRPM4 in the apo-closed, Ca/PIP₂-bound open, and Ca-bound closed states showed that the interaction between F910 on S4 and F935 on S5 plays the central role in coupling the Ca²⁺-induced conformational change at S1-S4 to the pore-opening. The close contact between F910 and F935 is well defined in the apo-closed and open structures but is absent in the Ca-bound closed structure. The trajectory of S4 movement upon Ca²⁺ binding in S1-S4 would directly push S5 through the F910/F935 contact and drive the pore to an open conformation. Thus, the loss of F910/F935 contact due to their side-chain movement in the Ca²⁺-bound structure likely happens after pore opening, which decouples the pore opening from Ca²⁺-induced S1-S4 conformational change and results in the closure of the pore. Of course, we cannot rule out the possibility that the Ca²⁺-bound structure may represent an intermediate state before channel opening and it will be difficult, if not impossible, to design a functional assay to distinguish this possibility from channel desensitization. The use of “intermediate conformation” to describe the Ca²⁺-bound structure in our initial version could also cause some confusion. In the revised manuscript, we have reworded part of the description regarding TRPM4 desensitization to clarify.

4) The authors claim that the ATP-bound state is an ATP-inactivated state, perhaps because the authors included both Ca²⁺, PIP₂, and ATP. However, it is possible that in most cases, ATP binds to the apo state first and then stabilizes the apo state in the sample. Consistent with this scenario, the MHR arrangement of the ATP-bound state looks very similar to the apo state, and the role of ATP seems to strengthen the inter-subunit interactions of MHRs (MHR1 and MHR3), as stated by the authors. In my opinion, the ATP-bound state represents the ATP-inhibited, and its role is to stabilize the closed state. For this reason, I suggest changing the definition of the state as “ATP-inhibited state” unless the authors have functional data to support that this conformation as an inactivated (deactivated? Why inactivated?) state.

We agree with this reviewer’s suggestion and have used the ATP-inhibited state to describe the ATP-bound structure in the revised manuscript.

5) Finally, I noticed that the current ATP-inactivated state lacks Ca²⁺ in S1-S4, which is interesting, while the previous study appears to have Ca²⁺ in S1-S4. Additionally, the previous study indicates a different ATP binding site at 37 °C compared to 22 °C. Please compare the current ATP-bound structure with the previously published structures in a revision.

In our study, ATP stabilizes the channel in a closed conformation, and we didn't observe bound Ca²⁺ within S1-S4. Also, because the intracellular domains remain in the closed conformation, the intracellular Ca²⁺ site cannot form as discussed in our earlier response. In our ATP-bound structure, the adenine base engages in most of the protein-ligand interactions with its surrounding aromatic residues from the MHR1 domain, and the phosphate groups interact with

residues (S422 and R421) of MHR3 from the neighboring subunit. In the previous ATP-bound structure determined at 37 °C, the pore remains closed but the entire intracellular domains undergo an upward swing movement and adopt a conformation somewhere between the open and closed states, likely caused by higher temperature. The adenine base maintains the same interactions with those aromatic residues from MHR1 (as shown in the figure beside). Because of the domain movement, ATP moves along with MHR1, making it appear to bind at a different site. This movement abolishes the interactions between the

ATP phosphate group and MHR3 domain observed in our closed structure. Also, because of the intracellular domain movement, the intracellular Ca²⁺ site appears to reform in the high-temperature structure. However, it is unclear what state this high-temperature ATP-bound structure represents. With very different conditions used to obtain the ATP-bound structures between ours and their study, it is difficult to interpret these structural differences and their physiological relevance. We therefore decided not to include the suggested comparison in the revision.

Minor points

1) Line 148: please provide references for TRPV1, TRPV5, TRPM3, and TRPM8 studies.

References added as suggested.

2) Line 155-156: Based on the structural analysis, the authors conclude that Ca²⁺ binds and activates first then PIP2 binds. Although it is logical based on the conformational selection mechanism, it is also possible that initial PIP2 binding can induce conformation for Ca²⁺ binding or synergistic Ca²⁺ and PIP2 binding (induced fit type). Also, it seems the authors purify protein with PIP2 and then add Ca²⁺, so PIP2 should bind to protein at least loosely first. Thus, I suggest the authors tone down this sentence, acknowledging the other possibilities unless the authors have functional data.

We agree with the reviewer's suggestion that PIP2 likely binds loosely to channel even in the absence of Ca²⁺ and it is possible that Ca²⁺ and PIP2 work synergistically to activate the channel. We have toned down this sentence as suggested. It is worth noting that PI(4,5)P2 cannot activate the channel by itself and we have never observed a TRPM4 structure with only PI(4,5)P2 bound.

Response to reviewer #1:

While the manuscript is an important contribution, I was somewhat disappointed that the authors did not put appropriate effort into performing a simple, but important requested experiment demonstrating the effect of native long acyl chain PIP₂ on TRPM4. While natural PIP₂ is somewhat more difficult to handle than DiC8 PIP₂, there are ample reports in the literature on PIP₂ and ion channels from many labs, including early papers the Hilgemann, Nichols and Logothetis labs showing clear effects of natural PIP₂ on ion channels in excised inside out patches. After sonication in an aqueous buffer, this lipid was shown to activate a wide variety of PIP₂ sensitive ion channels. The effect is usually slower than that of DiC8 PIP₂, but on most channels it requires lower concentrations and gives a larger effect than DiC8 PIP₂. I think this is a critical experiment, and the authors should put appropriate effort into performing it and including in the manuscript. Dipalmitoyl PIP₂ is an alternative, if natural PIP₂ remains too problematic, it is more soluble than native PIP₂, but it is a lot closer than DiC8 PIP₂ to the natural form.

We performed the experiment to demonstrate the activation effect of native long-chain PIP₂ on TRPM4. These new data are now included as **Supplementary Fig. 1**, along with a description in the functional characterization section. As noted previously, native PIP₂ is poorly soluble and tends to form liposomes, requiring extensive sonication and prolonged perfusion (several minutes) to observe potentiation. Furthermore, the high cytosolic [Ca²⁺] required for TRPM4 activation also stimulates phospholipase C-mediated PIP₂ hydrolysis, which counteracts the potentiation effect of native PIP₂.

The earlier studies cited by the reviewer likely refer to lipid activation of various inwardly rectifying potassium (Kir) channels. While native long-chain PIP₂ can strongly - but slowly, due to its poor solubility - activate Kir channels, TRPM4 (like other PIP₂-sensitive channels we have studied) responds equally well or more robustly to diC8-PIP₂. The enhanced response is attributed to diC8-PIP₂'s water solubility and rapid incorporation into the membrane. For this reason, diC8-PIP₂ is widely used as a substitute for native PIP₂ in electrophysiological studies of TRPM4 and related channels.

I also think the authors should discuss at least some of the structures on other TRP(M) channels and PIP₂. I do not believe it would be out of place. The current discussion of the manuscript is very short, and such discussion, even if brief, would make the manuscript more accessible to a wider readership.

As requested, we have incorporated a discussion of PIP₂ binding in the structures of other TRP channels (TRPM and TRPV) in the **Discussion** section of the revised manuscript.

Response to reviewer #2:

The revised manuscript has addressed most of my concerns and is improved. I appreciate the authors' structural analysis of the Ca²⁺-bound closed state and their rationale for designating it as a desensitized state. However, as the authors acknowledge, it remains equally plausible that this conformation represents an intermediate state rather than a desensitized one. Given that Ca²⁺ binding/activation initiates in the S1–S4 then propagates to the pore regions, the closed conformation of the MHR domain in the absence of bound Ca²⁺ is also consistent with an intermediate state. Alternatively, if this conformation reflects a desensitized state, the authors will need to invoke Ca²⁺ release from the MHRs after channel opening—an energetically challenging scenario that is not convincingly supported. Regardless, this is an important study, and I do not want to delay its publication. To proceed without further revision, I suggest that the authors discuss both possibilities—the intermediate and desensitized interpretations—and provide a rationale for favoring the desensitized state. Importantly, the authors should refer to this conformation as a putative desensitized state throughout the manuscript, which would be more accurate based on the current data. Again, congratulations on this milestone study on the mechanism of TRPM4 activation.

We appreciate the reviewer's supportive feedback. While we hold a different perspective on the classification of the Ca²⁺-bound TRPM4 structure, we agree that we should not exclude the possibility of it being in an intermediate state. In line with the suggestion, we have revised the description of the Ca²⁺-bound structure to reflect both interpretations and included our rationale for designating it as a desensitized state. Accordingly, we now refer to the Ca²⁺-bound structure as the *putative desensitized state* in the revised manuscript, as suggested.